# Validation of the Aeolus Level-2B wind product over Northern Canada and the Arctic

Chih-Chun Chou[1], Paul J. Kushner[1], Stéphane Laroche[2], Zen Mariani[2], Peter Rodriguez[2], Stella Melo[2], Christopher G. Fletcher[3]

[1]Department of Physics, University of Toronto, Toronto, M5S 1A7, Canada
[2]Environment and Climate Change Canada.
[3]Department of Geography and Environmental Management, University of Waterloo, Canada.

*Correspondence to*: Chih-Chun Chou (gina.chou@mail.utoronto.ca)

**Abstract.** In August 2018, the European Space Agency launched the Aeolus satellite, whose Atmospheric LAser Doppler INstrument (ALADIN) is the first spaceborne Doppler wind lidar to regularly measure vertical profiles of horizontal line-of-sight (HLOS) winds with global sampling. This mission is intended to assess improvement to numerical weather prediction provided by wind observations in regions poorly constrained by atmospheric mass, such as the tropics, but also, potentially, in polar regions such as the Arctic where direct wind observations are especially sparse. There remain gaps in the evaluation of

the Aeolus products over the Arctic region, which is the focus of this contribution. Here, an assessment of the Aeolus Level-2B wind product is carried out progressing from specific locations in the Canadian North, to the pan-Arctic. In particular, Aeolus data is compared to a limited sample of coincident ground-based Ka-band radar measurements at Iqaluit, Nunavut; to a larger set of coincident radiosonde measurements over the Canadian North; to Environment and Climate Change Canada (ECCC)'s short-range forecast; and to the reanalysis product, ERA5, from the European Centre for Medium-Range Weather

Forecasts (ECMWF). Periods covered include the early phase of the first laser flight model (FM-A; 2018-09 to 2018-10), the early phase of the second laser flight model (FM-B; 2019-08 to 2019-09), and the middle phase of FM-B (2019-12 to 2020-01). The adjusted r-squared between Aeolus and other local datasets are around 0.9 except for lower values for the comparison with the Ka-band radar, reflecting limited sampling opportunities with the radar data. This consistency is degraded by about 10% for the Rayleigh winds in the summer due to solar background noise and other possible errors. Over the pan-Arctic,

consistency, with correlation greater than 0.8, is found in the Mie channel from the planetary boundary layer to the lower stratosphere (near surface to 16 km a.g.l.) and in the Rayleigh channel from the troposphere to the stratosphere (2 km to 25 km a.g.l.). In all three periods, Aeolus standard deviations are found to be 5 to 40% greater than those from ECCC-B and ERA5. We found that the L2B estimated error product for Aeolus is coherent with the differences between Aeolus and the other datasets, and can be used as a guide for expected consistency. Our work shows that the high quality of the Aeolus dataset that

has been demonstrated globally applies to the sparsely sampled Arctic region. It also demonstrates the lack of available independent wind measurements in the Canadian North, lending urgency to the need to augment the observing capacity in this region to ensure suitable calibration and validation of future space-borne DWL missions.

# 1 Introduction

A better characterization of the global wind field has the potential to improve numerical weather prediction (NWP) and thereby improve our knowledge of the transport of moisture, energy, and other fields in the global atmosphere (Baker et al.,1995; Graham et al., 2000, Naakka et al., 2019). Altitude-resolved wind observations are available from aircraft reports and surface-based observations such as radiosondes and wind profilers. However, such observations are generally scattered and especially rare over large bodies of water such as the world's oceans, as well as over polar regions. Winds derived from passive space-based observations include atmospheric motion vectors (AMVs) estimated from the movements of clouds and water vapour (Velden et al., 2017; Mizyak et al., 2016) and surface winds from space-based scatterometer from the ocean surface. Although AMV products provide wind information over multiple tropospheric layers using multispectral water vapor remote sensing (Velden et al., 1997; Bormann and Thépaut, 2004; Le Marshall et al., 2008), they lack precision in terms of altitude assignment and their sampling is limited to only a few levels. This limits how AMV's represent the small-scale vertical structure of the wind profile. Spaceborne scatterometers, on the other hand, are limited to ocean near-surface winds and their accuracy is therefore sensitive to surface weather conditions (Chiara et al., 2017; Young et al., 2017). Improving altitude-resolved winds from remote sensing on a global scale requires adoption of active sensors, which have only recently become feasible for deployment from space-based platforms (Dabas, 2010).

On 22 August 2018, the European Space Agency (ESA) launched the Aeolus satellite carrying the first spaceborne Doppler wind lidar (DWL) designed to significantly improve altitude-resolved wind observations, from the surface to the stratosphere, on a global scale (Källen, 2018; Reitebuch et al., 2019). The instrument carries an emitting UV laser and two receivers to measure the Doppler shift from backscattering by air molecules (Rayleigh channel) and by aerosols or cloud particles (Mie channel). Aeolus was designed to improve global weather forecasts, with an emphasis on tropical winds, because tropical wind information is required to fully characterize the circulation when dynamical balance constraints are weak (Horányi et al., 2015). However, since it is polar orbiting, Aeolus also fills an observation gap in the polar regions, including the Arctic region, which is our focus. It is worthwhile exploring how new measurements of Arctic winds, along with other meteorological observations, might improve Arctic forecasts (e.g., Yamazaki et al., 2015), and, by extension, prediction outside the Arctic (Naakka et al., 2019; Lawrence et al., 2019), with a potential to influence forecasting and characterization of mid-latitude weather and climate extremes (Walsh et al., 2019; Cohen et al., 2020; Sato et al., 2017). We are thus motivated to better understand the quality of Aeolus data products in the Arctic region, particularly for Canada, given its large territorial extent at high northern latitudes.

The purpose of this paper is to evaluate the quality of Aeolus wind products over the Canadian North and the Arctic in comparison with several available observational products. We will focus on analyzing random errors instead of systematic errors since, as recommended for operational NWP practice, bias corrected Aeolus data is used in this study (see Sect. 2.1).

The products compared include the dataset from the Canadian Arctic Weather Science (CAWS) project, which includes ground based remote sensing and in-situ instruments for enhanced meteorological observations. As part of the Canadian contribution to the international calibration/validation effort for Aeolus (Martin et al., 2021; Guo et al., 2020; Baars et al., 2020), this project serves to test the spaceborne DWL that provides alternative observational wind data to atmospheric monitoring over the northern regions. The CAWS "supersites" are located at Iqaluit, NU (64° N, 69° W) and Whitehorse, YK (61° N, 135° W) (Joe et al., 2020), but because of data limitations (see below), only Iqaluit ground-based remote sensing data, along with Whitehorse radiosonde data, will be used in this study.

In related Arctic-based work, Belova et al. (2021) have found consistency between Aeolus winds and a ground-based radar situated in northern Sweden with insignificant biases between the two products (less than 1 ms$^{-1}$) and slightly increased random errors for Aeolus in the boreal summer, possibly due to sunlight scatter. We here build on this encouraging study by moving from a narrow focus at Iqaluit and Whitehorse, motivated by CAWS; to a broader set of radiosonde network locations across the Canadian North, including Iqaluit and Whitehorse; and finally, to a pan-Arctic perspective. Products compared to the Aeolus wind products include the Iqaluit Ka-band radar data; radiosonde data across the Canadian North, including Iqaluit and Whitehorse, and global data-assimilation based wind products, including the short-range forecast from ECCC's operational NWP system (ECCC-B) and the fifth major global reanalysis of the European Centre for Medium-Range Weather Forecasts (ECMWF ERA5). Section 2 provides a description of each of these datasets. Section 3.1 describes the validation against Iqaluit and the other Canadian Arctic sites, and Sect. 3.2 describes a broader validation for the pan-Arctic against background and reanalysis products. A summary and discussion of the results is provided in Sect. 4.

## 2 Datasets

We now discuss the Aeolus wind products (Sect. 2.1), the other datasets that will be compared with the Aeolus wind products (Sects. 2.2-2.4), and the data matching process including coincidence criteria used in the validation (Sect. 2.5).

### 2.1 Aeolus L2B HLOS wind product

The near polar-orbiting and sun-synchronous Aeolus satellite measures global atmospheric wind profiles along the DWL's line-of-sight (LOS) from the Earth's surface to the lower stratosphere (Straume et al., 2018). The LOS of Aeolus is perpendicular to its orbital velocity to mitigate contributions from its along-orbit velocity. Its DWL, the Atmospheric LAser Doppler INstrument (ALADIN, Guo et al., 2020), points 35° from the nadir and includes two receivers to measure the Doppler shift from the emitting laser along the LOS: a double Fabry-Pérot spectrometer to measure Rayleigh scattering from air molecules and a Fizeau spectrometer to measure Mie scattering from cloud droplets and aerosols. The horizontal line-of-sight (HLOS) wind component can be derived by analyzing the Doppler frequency shift and assuming that the vertical component of winds is negligible. The wind retrieval method of the processed and calibrated Aeolus Level-2B (L2B) HLOS wind product

can be found in the Algorithm Theoretical Basis Documents (Rennie et al., 2020a). For both Mie and Rayleigh channels each measurement bin is classified into "cloudy" or "clear" using its optical property information from Level-1B scattering ratio estimates (Rennie et al., 2020a). "Cloudy" classification occurs when the measurement-bins have non-zero particle backscatter, while "clear" classification occurs for predominantly molecular backscatter. Since Mie-cloudy and Rayleigh-clear winds are considered as superior quality compared to Mie-clear and Rayleigh-cloudy winds (Martin et al., 2021; Guo et al., 2020; Baars et al., 2020), Mie winds and Rayleigh winds refer exclusively to Mie-cloudy and Rayleigh-clear winds in the rest of this study.

The backscattered signal must be horizontally and vertically averaged to obtain a sufficient signal-to-noise ratio (SNR) (Drinkwater et al., 2016; Reitebuch et al., 2019; Lux et al., 2020). Prior to 5 March 2019, both Rayleigh and Mie winds were averaged to up to a horizontal resolution of 87 km. Recognizing that Mie scattering in cloudy air yields stronger returns than Rayleigh scattering in clear air, after 5 March 2019, the Mie wind product was provided at a finer horizontal resolution of 12 km. The vertical resolution decreases from 500 m in the PBL (defined here as below 2 km in altitude) to 1 km in the free troposphere (defined here as 2 to 16 km in altitude) and to 2 km in the lower stratosphere (above 16 km in altitude). The Mie channel covers the vertical range up to 16 km in altitude and the Rayleigh channel covers up to 30 km.

Aeolus switched from the first laser, flight model A (FM-A) to the second laser, flight model B (FM-B) due to a decrease in ultraviolet (UV) power output from FM-A at the end of June 2019 (Reitebuch et al., 2019; Lux et al., 2020). Aeolus L2B near real-time baseline products 2B02 and 2B06/07 are used during early FM-A period (15 September to 16 October 2018) and FM-B period (2 August to 30 September 2019 and 1 December 2019 to 31 January 2020) respectively. ECMWF has published the first reprocessed data in fall 2020 (2B10; available at ftp://2018_aeolus_l2b:ecmwf@acquisition.ecmwf.int/), which covers the period between 24 June and 31 December 2019. The major improvement in this product is a daily updated bias correction accounting for variability of the temperature gradients across the detector telescope's primary M1 mirror; additional improvements are mentioned below and in other studies (e.g., Rennie and Isaksen, 2020; Laroche and St. James, 2021). A comparison of the statistical results during the overlapping period, (boreal) summer 2019, between 2B06 and 2B10 will be presented in this study.

The following data selection is carried out in this study:

- L2B product provides a validation flag of 1 (valid) or 0 (invalid) (de Kloe et al., 2016) associated with each range-bin in an observation, and we therefore screen out validation flag value 0 (Baars et al., 2020).
- The quality control recommendation following Rennie and Isaksen, (2020). The thresholds for L2B estimated observation errors during the FM-A period are 4.5 ms$^{-1}$ for the Mie winds and 6.6 to 11 ms$^{-1}$ for the Rayleigh winds, depending on the pressure level, and 5 ms$^{-1}$ for the Mie winds and 8.5 to 12 ms$^{-1}$ for the Rayleigh winds during the early FM-B period. For more details, please refer to Rennie and Isaksen (2020).

- We further reject the outliers by excluding data for which the difference between the observations and ECCC-B or ERA5 is greater than 30 ms$^{-1}$. This criterion was obtained from an initial comparison between Aeolus FM-A and ECCC-B (Laroche et al., 2019). The outliers represent less than 1% of all data; however, excluding them has an important influence because their magnitude could be as large as 150 ms$^{-1}$.

During the early FM-A period, a global constant bias offset of -1.35 ms$^{-1}$ was added to the Mie winds to bring them into better agreement with the ECMWF model (Rennie and Isaksen, 2020). The Aeolus observation heights were also systematically increased by 250 m due to a known calibration issue. The biases of FM-B HLOS arising mainly from the telescope primary mirror M1 temperature gradients (Rennie and Isaksen, 2020) should be corrected as much as possible before any use for validation against other wind measurements or data assimilation. Fortunately, these biases vary mostly with the orbital node and latitude and partly with longitude and height, facilitating such bias correction. ECCC has developed a bias correction scheme similar to ECMWF, as described in Rennie and Isaksen (2020); see Laroche and St. James (2021). It is a look-up table bias correction based on the mean observation minus the "background" short-range forecast from ECCC (see Sect. 2.2) from the previous 7 days, as a function of orbit phase and latitude. It is applied for both Rayleigh and Mie HLOS winds. For the Rayleigh HLOS winds, the correction is also a function of longitude, binned in 10 degrees latitude by 36-degree longitude sectors.

To project the wind vector in a given dataset into the Aeolus HLOS, we use

$$v_{HLOS} = -u \sin \varphi - v \cos \varphi, \tag{1}$$

where $v_{HLOS}$ is the HLOS wind component, $u$ is the zonal wind component, $v$ is the meridional wind component, and $\varphi$ is the azimuth of the LOS. Conversely, we can also project the HLOS wind vector into the west-east and north-south directions (Wright et al., 2021) for some analysis (Sect. 3.2), using

$$v_{HLOS,u} = -v_{HLOS} \cdot \sin (\varphi), \tag{2}$$

$$v_{HLOS,v} = -v_{HLOS} \cdot \cos (\varphi). \tag{3}$$

To repeat, these quantities do not represent zonal and meridional components of the total wind field, but the zonal and meridional projection of the vector component of the wind along the HLOS of Aeolus.

## 2.2 ECCC-B: Short-range forecast (background) from ECCC

The "background" from ECCC, termed "ECCC-B", is the 9-h short-range forecast used in the operational four-dimensional ensemble-variational (4D-EnVar) data assimilation scheme (Buehner et al., 2015). The forecast model is the operational Global Environmental Multiscale (GEM) (McTaggart-Cowan et al., 2019) with 15 km horizontal grid spacing and 84 vertical levels. There are over 13 million observations assimilated daily during the periods examined in this study, which include data from

infrared (56.1% of all observations assimilated) and microwave (27.7%) satellite sounders and imagers, aircraft (9.6%), atmospheric motion vectors (2.3%), radiosondes (2.1%), scatterometers (1.0%), near-surface observations (0.7%), and satellite-based radio occultation (0.4%). For the comparison between ECCC-B and Aeolus winds, the closest short-range forecast field, available every 15 minutes, is selected. Then, this field is linearly interpolated in space to Aeolus measurement locations, first horizontally and then vertically. For the linear interpolation between the model's grid points, the horizontal grid-spacing is 15 km and the vertical grid-spacing varies from approximately 100 m in the PBL to 1 km in the stratosphere (McTaggart-Cowan et al., 2019). The linear interpolation in time is between two consecutive model states, 15 min apart.

## 2.3 Reanalysis ERA5

The ERA5 hourly data on 37 pressure levels from the ECMWF is also used in this study in validating the Aeolus measurements. This dataset is based on a four-dimensional variational (4DVar) data assimilation using Cycle 41r2 of the IFS, which was introduced operationally in 2016. ERA5 provides hourly estimates of atmospheric, land, and oceanic climate variables, available from 1950 to present. Data is gridded in a regular latitude-longitude grid of 0.25 degrees. A further discussion of the ERA5 configuration can be found in Hersbach et al. (2018 and 2020). The process used to match ERA5 and Aeolus data will be discussed in Sect. 2.5.

## 2.4 Ground-based measurements at Iqaluit, Nunavut; Whitehorse, Yukon; and other radiosonde stations

The CAWS project, led by ECCC, aims to characterize and improve scientific understanding of Arctic weather, climate, and cryospheric systems through enhanced meteorological observation capacity (Joe et al., 2020; Mariani et al., 2018). It also seeks to improve weather forecasts in the Canadian Arctic, test new technologies, and calibrate and validate space-based observations. ECCC's Iqaluit and Whitehorse sites (Fig. 1), so-called "supersites", were identified as "hot spots" for both extreme weather and transportation infrastructure that merited additional instrumentation. They provide researchers and forecasters with real-time weather observations which can be used in evaluating NWP models. Connected to ECCC's observational science mission, locating these weather stations at high latitudes also tests the ability of the coordinated instrument suites to operate in extreme cold conditions.

The Iqaluit site is situated in a valley to the north-east overlooking Frobisher Bay in the vicinity of 300 m hills. We focus on two instruments at Iqaluit site that provide wind profiles measurements: the radiosonde and Ka-band radar. We will also briefly mention ground-based Doppler lidar measurements in Sect. 3.1. Vaisala RS92 radiosondes (Mariani et al., 2018) were launched twice daily (45 minutes before synoptic times 00 and 12 Coordinated Universal Time (UTC)). They measure vector wind profiles with a vertical resolution of roughly 15 m depending on ascent speed, up to about 30 km above ground level. The data used (available at http://weather.uwyo.edu/upperair/sounding.html) is the processed radiosonde data provided

at mandatory and significant pressure levels (which has a coarser resolution than 15 m). It takes about two hours to reach 30 km altitude (around 10 hPa). The instrumental uncertainty for the wind speed is between 0.4 and 1.0 ms$^{-1}$ and between 0.3 and

0.7 ms$^{-1}$ for the zonal wind component (Dirksen et al., 2014). The error on the zonal wind component due to drift and elapsed time of the ascending balloon is between 0.5 and 1.0 ms$^{-1}$ in the troposphere and upper troposphere/lower stratosphere (UTLS) (see Fig. 5b in Laroche and Sarrazin, 2013). As a result, the total error for the zonal wind component from these sources of errors is between 0.6 and 1.2 ms$^{-1}$. Note that the radiosonde data are assimilated in the ECCC and ECMWF systems, which means that the ECCC-B and ERA5 errors are not independent of the radiosonde observation errors. The ECCC Whitehorse

site, situated in a wide valley with large lakes, also has radiosondes that operate similarly to the ones at the Iqaluit.

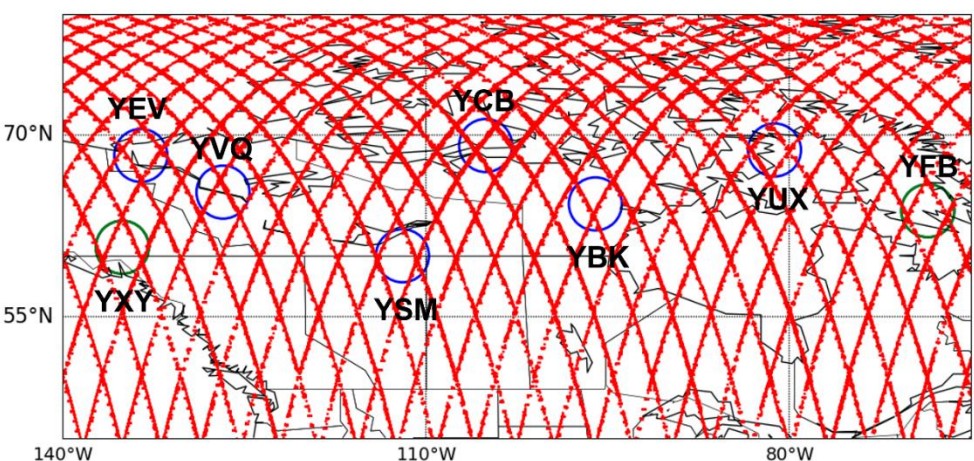

**Figure 1.** Aeolus's overpasses centred over the Canadian North (red dots) during the first week of August 2019. The green 90-km radius circles centred on Iqaluit (YFB) and Whitehorse (YXY) within which coincident Aeolus overpasses were compared with other datasets. The blue circles indicate the locations of other radiosonde stations over the Canadian Arctic: Inuvik (YEV), Fort Smith (YSM), Hall Beach (YUX), Cambridge Bay (YCB), Norman Wells Ua (YVQ), and Baker Lake (YBK).


       The dual-polarization Doppler Ka-band radar at Iqaluit measures the LOS wind speed, fog backscatter, and depolarization ratio every 15 minutes. The radar measures the LOS wind with 14 m resolution and the LOS range goes from 5 to 30 km, depending on hydrometeor concentration. The uncertainty of the measurements depends on conditions, SNR, and decibel of the return signal. The average vertical wind profile bias to radiosonde is better than 0.3 ms$^{-1}$. The horizontal winds

are derived using a high angle plan position indicator (PPI) 75 degrees scan using the VAD (Velocity-Azimuth-Display) algorithm (Lhermitte and Atlas, 1962; Wang et al., 2010) whereby the radar scans with a fixed elevation angle ($\varphi$) while the azimuth angle ($\theta$) is varied. The radial velocity is given by

$$v_r = u \sin\theta \cos\varphi + v \cos\theta \cos\varphi + w \sin\varphi, \qquad\qquad (4)$$

where $w$ is the vertical wind component. By fitting the data and assuming uniform winds at each range, these three unknown

parameters ($u$, $v$, and $w$) can be derived at each vertical level.

Other than the ECCC supersites, we also validate the Aeolus wind product in comparison with radiosonde measurements over the Canadian Arctic at ground stations in Inuvik, Fort Smith, Hall Beach, Cambridge Bay, Norman Wells Ua, and Baker Lake (Fig. 1). They operate similarly to the radiosondes at Iqaluit and Whitehorse and measure vector wind profiles. Some of the stations launch the radiosondes four times a day at 00, 06, 12, and 18 UTC. However, this does not affect
the temporal criteria (see Sect. 2.5).

## 2.5 Data matching process and coincidence criteria

For the ground-based validation, the criterion for coincidence of Aeolus overpasses is that the distance from the sites to the measurements be no more than 90 km (horizontal resolution of Rayleigh winds). Using this coincidence criterion, Aeolus overpasses are selected as targets for validation at Iqaluit three times a week at around 21:50, 11:15, and 22:00 UTC, and at
Whitehorse twice a week at around 02:25 and 15:30 UTC. The Aeolus measurements are compared to the reanalysis and in-situ measurements that are available in the nearest time. Temporal sampling for each product is as follows: Aeolus overpasses at Iqaluit and Whitehorse are as mentioned above; reanalysis data is provided hourly, on the hour; radiosonde data is from launches at 00 and 12 UTC, with a two-hour time-of-flight to 30 km as mentioned above; Ka-band radar data is provided via 15-minute scans. For example, if Aeolus overpasses selected as a target for validation at the Iqaluit site occur at 11:15 UTC,
the Aeolus HLOS profile would be compared to the reanalysis data at 11 UTC, to radiosonde measurements at 12 UTC, and to the nearest scan by the radar. On the other hand, if the overpass time is 02:25 UTC, the profile would be compared to the ERA5 data at 02 UTC, the radiosonde measurements at 00 UTC, and, again, the nearest scan by the radar.

## 3 Results

### 3.1 Validation against ground-based measurements in the Canadian Arctic

We evaluated the vertical HLOS wind profile observations from coincident Aeolus overpasses for Iqaluit and Whitehorse against ground-based measurements, ECCC-B, and reanalysis. Our evaluation was limited to the early FM-A period of Aeolus because the Ka-band radar at Iqaluit has been turned off for repairs since 1 August 2019. Figure 2 shows examples of wind profile measurements on (a) 22 September 2018 when Aeolus was in its ascending orbit phase and (b) 24 September 2018 when Aeolus was in its descending orbit phase, at the Iqaluit site. The HLOS wind profile is shown, along

with profiles of the zonal projection of the HLOS component (dashed curves), $v_{HLOS,u}$ from equation (2), for ERA5, radiosonde, Ka-band radar, and lidar. When the measured winds are positive, it means the HLOS winds are directed away from the instrument (eastward for the ascending orbit phase and westward for the descending orbit phase). To ease the interpretation, we plot the negative HLOS winds when Aeolus is in its descending orbit phase (panel b). The Ka-band radar's vertical range extends to less than 5 km in both profiles, around where there are Mie wind measurements from Aeolus, because

its vertical range depends on hydrometeor concentration, and its sampling of coincident timing is limited for the Aeolus measurement period. The Iqaluit site also hosts a ground-based Doppler lidar whose LOS wind measurements yields horizontal vector winds via VAD up to about 3 km a.g.l. or the cloud base height. This instrument provides very limited coincident measurement opportunities with Aeolus due to its limited vertical range. Observations from this instrument, which have been extensively validated against high-resolution radiosondes (Mariani et al., 2020) and are useful for boundary-layer focused

work, are also shown in Fig. 2 for visual comparison only for 22 and 24 September.

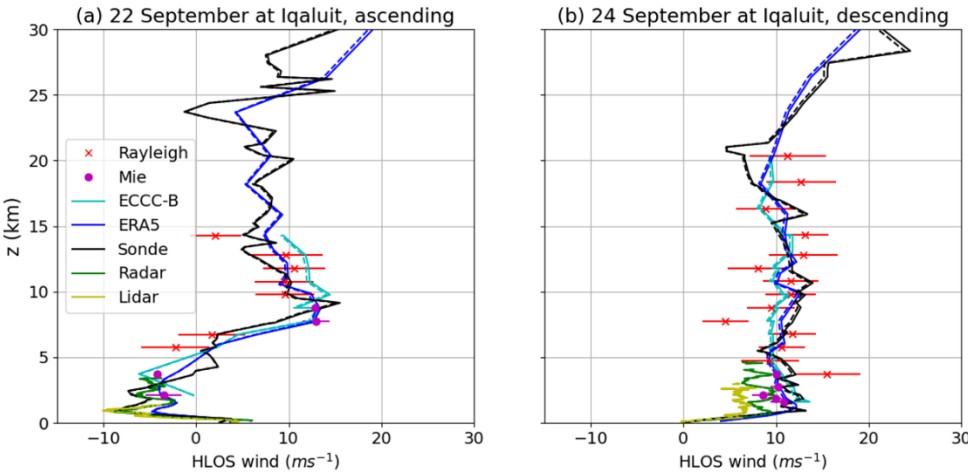

**Figure 2.** HLOS wind profile observations from coincident Aeolus overpasses (Rayleigh and Mie winds, along with L2B estimated error, i.e., wind error quantifier for each observation); ECCC-B, i.e., the short-range forecast (background) from the ECCC numerical weather prediction model; ERA5; and ground-based remote sensing observations (radiosonde, Ka-band radar, and lidar measurements), on (a) 22 September and (b) 24 September 2018. Also shown are zonal projection of the HLOS winds (dashed line). The HLOS winds are plotted so

that their zonal projection is positive eastward.

In Fig. 2, we see that Aeolus consistently captures some of the basic structure of the wind profiles compared to in-situ measurements, ERA5 reanalysis, and ECCC-B. Because the solid lines lie very close to the dashed lines, it is evident that Aeolus is providing predominantly zonal wind information even at high latitudes (63° N) where the LOS has a greater meridional component than at low latitudes. On 22 September, Aeolus detects an easterly wind feature in the lower atmosphere and accurately picks up the change of sign around 5 km altitude. On 24 September, although a few of the Rayleigh measurements have a deviation close to 50% from the other datasets around 8 km a.g.l., Aeolus still measures westerly winds in reasonable overall agreement with the other data.

Although the Ka band radar offers limited sampling, it is retained in this analysis because it offers an entirely independent and unique set of observations in the Canadian North that are not assimilated in any NWP model. Furthermore, it provides consistent measurements with the radiosondes, as shown in Fig. 3. For the same period of analysis and when the radar observations are within 30 minutes of radiosonde launch, the bias of the wind speed between the radar and radiosonde is less than 1 ms$^{-1}$ for measurements above 200 m a.g.l., and the standard deviation of the differences are within 3 ms$^{-1}$.

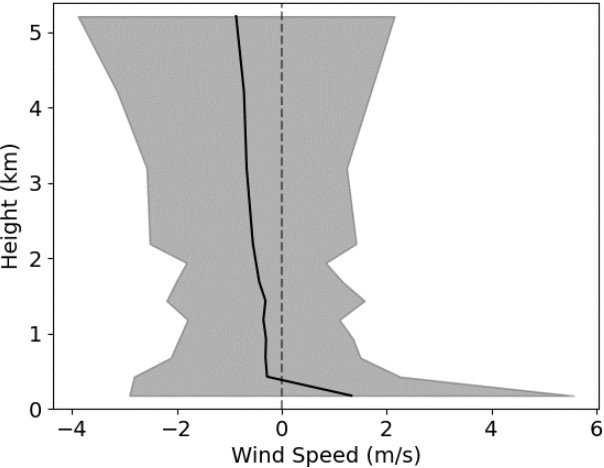

**Figure 3.** Iqaluit Ka-band radar bias (black line) and standard deviation of the differences (STDE, shaded region) compared to radiosonde measurements for coincident observations from 15 September to 16 October 2018. Only radar observations within 30 minutes of radiosonde launch were analyzed.

Figures 4 and 5 show scatter plots between the different datasets with lines of best fit and their range, and frequency distributions in percentage around Iqaluit (black) and Whitehorse (blue) sites. Figure 4 compares Rayleigh winds against the other products (ECCC-B, ERA5, radiosonde, and the limited number of coincident Ka-band radar profiles) and Fig. 5 compares Mie winds against the other products. Aeolus provides more Rayleigh measurements than Mie winds, because the Rayleigh channel measures winds under clear-sky conditions and has greater vertical extent, while the Mie channel measures winds under cloudy or high-aerosol conditions.

To measure the consistency of vertical profiles, we calculate the adjusted r-squared statistic, $r_{adj}^2$, using

$$r^2 = 1 - \frac{\Sigma_i(y_i - \hat{y}_i)^2}{\Sigma_i(y_i - \bar{y})^2}, \tag{5}$$

$$r_{adj}^2 = 1 - \frac{(1-r^2)(N-1)}{N-p-1}, \tag{6}$$

where $y_i$ is the Aeolus measurements (or other dataset shown on the y-axis), $\hat{y}_i$ is the estimated HLOS wind using linear regression, $\bar{y}$ is the mean of $y_i$, $N$ is the total number of measurements, and $p$ is the number of profiles. The adjustment avoids overestimating the raw correlation from the scatterplots by accounting for within-profile agreement. The $N$ and $p$ to calculate

the adjusted r-squared in Figs. 4-5 are shown in Table S1. The $r_{adj}^2$ and slope of the fitted line are shown in Table 1. Along with the consistency between ECCC-B and ERA5, these two datasets are also consistent with the radiosonde data as expected, because radiosonde measurements are used in the operational ECCC and ECMWF data assimilation systems. All adjusted r-squared values in this comparison are above 0.95 for both sites and the slopes of the fitted line are all $1 \pm 0.1$.

| | $r_{adj}^2$ | Slope |
|---|---|---|
| Aeolus Rayleigh vs. ECCC-B | 0.92 (0.95) | 1.05 (0.96) |
| Aeolus Rayleigh vs. ERA5 | 0.91 (0.95) | 1.05 (0.97) |
| Aeolus Rayleigh vs. radiosondes | 0.90 (0.89) | 1.02 (0.92) |
| Aeolus Rayleigh vs. radar | 0.53 | 1.01 |
| Aeolus Mie vs. ECCC-B | 0.87 (0.98) | 1.05 (0.98) |
| Aeolus Mie vs. ERA5 | 0.81 (0.99) | 1.02 (1.01) |
| Aeolus Mie vs. radiosondes | 0.82 (0.99) | 1.01 (1.01) |
| Aeolus Mie vs. radar | 0.66 | 1.00 |
| ECCC-B vs. ERA5 | 0.97 (0.99) | 0.99 (1.01) |
| ECCC-B vs. radiosondes | 0.95 (0.95) | 0.96 (0.98) |
| ERA5 vs. radiosondes | 0.98 (0.97) | 0.97 (0.98) |

**Table 1.** Information on the adjusted r-squared and slope of fitted line from Figs. 4-5 as well as the comparisons between ECCC-B, ERA5,
and radiosondes at Iqaluit, with values for Whitehorse shown in parentheses.

Overall, the datasets show strong consistency. ECCC-B and ERA5 are highly mutually consistent (Table 1; with adjusted r-squared greater than 0.97) and therefore show similar consistency with Aeolus (Figs. 4a-b and 5a-b). It can be seen that Aeolus winds are in general less consistent with ECCC-B, ERA5, and radiosondes at Iqaluit than the corresponding
observations at Whitehorse. Moreover, at Iqaluit, Rayleigh winds show a higher consistency than Mie winds, while the opposite is true for Whitehorse. One possible reason for this relates to the fact that the Mie channel samples winds in the lower atmosphere where winds are harder to assimilate or measure due to topography. Since Iqaluit is situated in tundra valleys with rocky outcrops that can cause increased variability in the wind field while Whitehorse is situated in large valleys with less wind variability due to topography, terrain effects might account for the difference in consistency. In addition, the overall
range extent of the HLOS wind samples is between -25 to 25 ms$^{-1}$ at Iqaluit and -45 to 45 ms$^{-1}$ at Whitehorse and r-squared is sensitive to the range of data (note the denominator of the second term in Eq. (5)). Overall, Aeolus data show good agreement with these three datasets with adjusted r-squared greater than 0.8.

On the other hand, the adjusted r-squared between Aeolus winds and Ka-band radar at Iqaluit are only 0.53 for Rayleigh winds and 0.66 for Mie winds. As mentioned above, this might reflect a sampling bias because the vertical range of 305 the instrument is relatively limited due to requirement for hydrometeors to be present, so that there are therefore relatively few points to sample. In addition, at larger ranges, the radar measures winds further from the radar and so the radar's measurement covers a larger volume. The validity of the assumption of uniform winds for the VAD calculation to be correct becomes less accurate as the range increases. However, we are comparing the VAD wind profile to a large distance along the track (87 km) as well, so this might not be the main cause. Terrain effects at Iqaluit, mentioned above, might be another possible issue: 310 Aeolus might average out wind variability due to the topography over the resolution provided by the HLOS products. This reflects the challenge involved when comparing measurements with distinct spatial sampling and limited coincidence. Generally, the sampling for these radar measurements is highly limited, which tends to reduce the agreement compared to the other datasets. Nevertheless, the agreement on the variances between Aeolus and the Ka-band radar is at 99% confidence level using F-test. This analysis highlights the importance for programs such as CAWS to continue to provide ground-based radar 315 measurements to ensure independent measurements of the winds for future DWL missions.

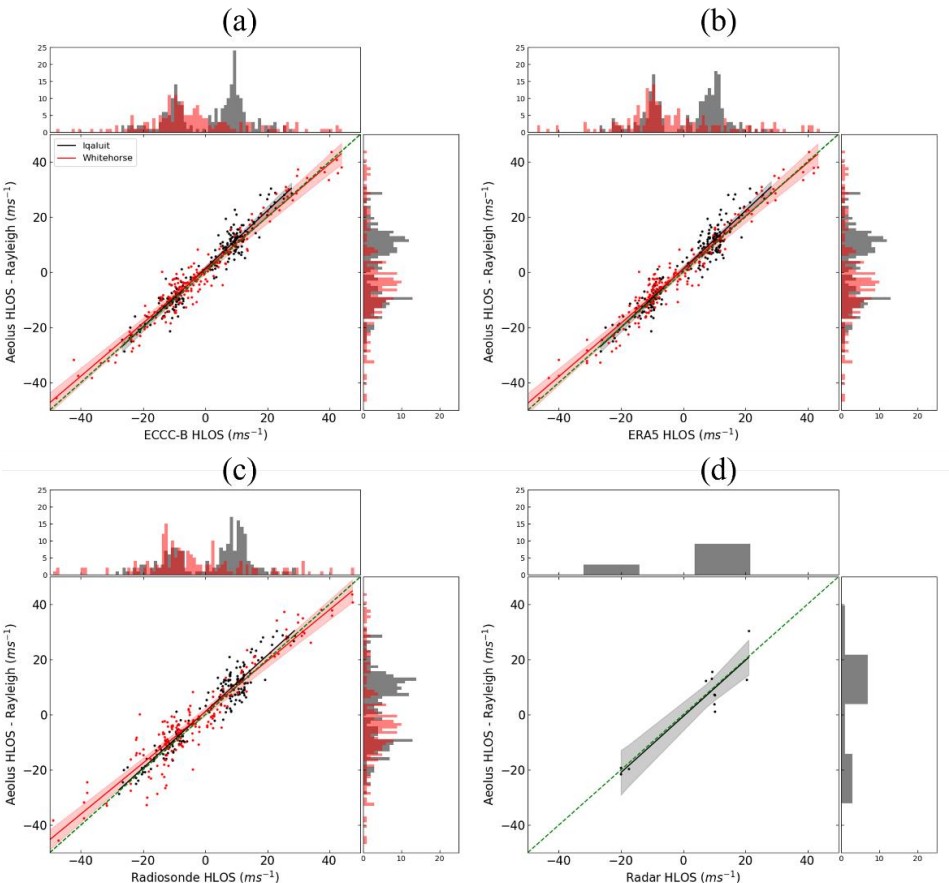

**Figure 4.** Scatter plots between Aeolus Rayleigh winds and (a) ECCC-B, (b) ERA5, (c) radiosondes, and (d) Ka-band Radar and frequency distributions in percentage around Iqaluit and Whitehorse supersites during the early FM-A period.

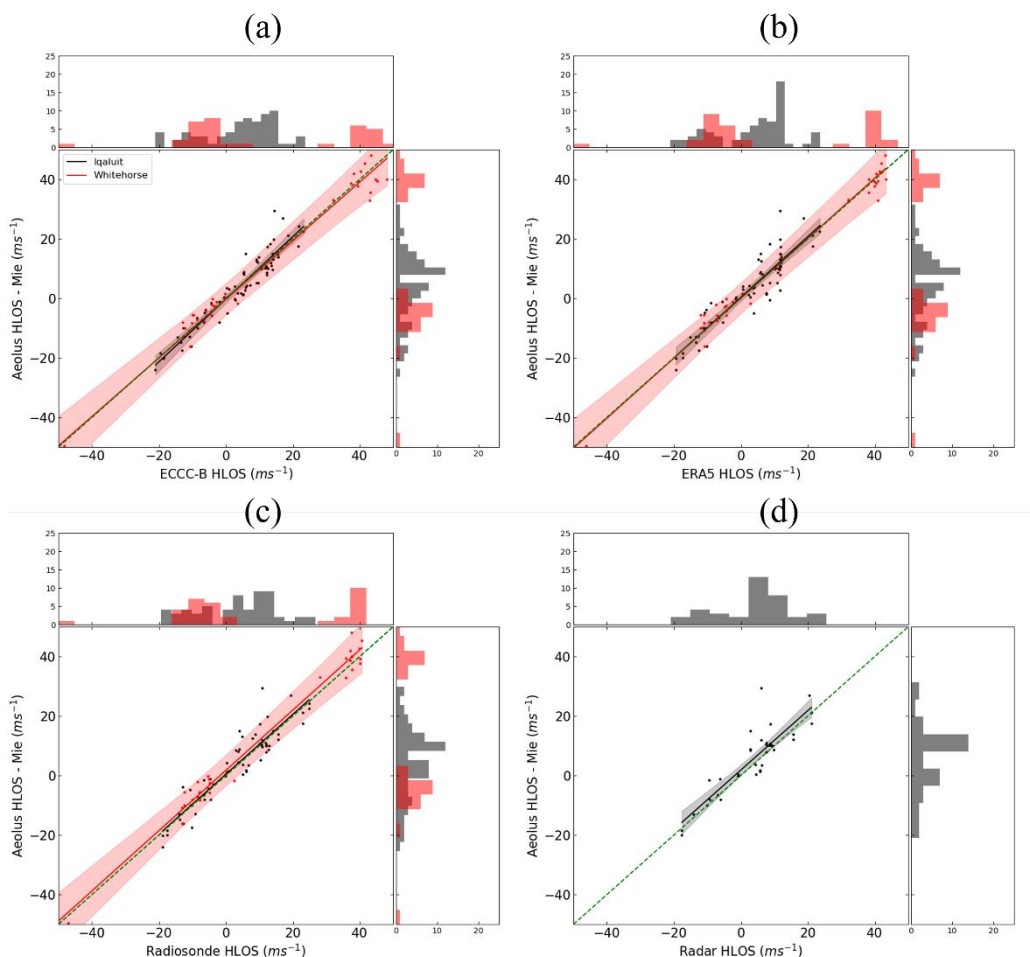

**Figure 5.** Similar to Figure 4, but scatter plots between Aeolus Mie winds and other datasets.


We broaden the region of analysis to the Canadian Arctic by incorporating all available Canadian Arctic radiosonde stations that provide wind profile observations. Figure 6 shows a comparison of adjusted r-squared ($N$ and $p$ are shown in Table S2) between 2B02/06/07 Aeolus and ECCC-B, ERA5, and radiosonde measurements coincident with the radiosonde stations shown in Fig. 1, during early FM-A, early FM-B, and mid-FM-B periods. Aeolus wind profiles are less consistent

with radiosonde measurements than with ECCC-B and ERA5 for both Rayleigh and Mie winds during all three periods of analysis (adjusted r-squared values in the range 0.01-0.05 lower for the radiosonde observations). These results are in good agreement with those of Martin et al. (2020), who showed that the representativeness error is significantly larger for radiosonde observations than those for the ECMWF and ICON models.

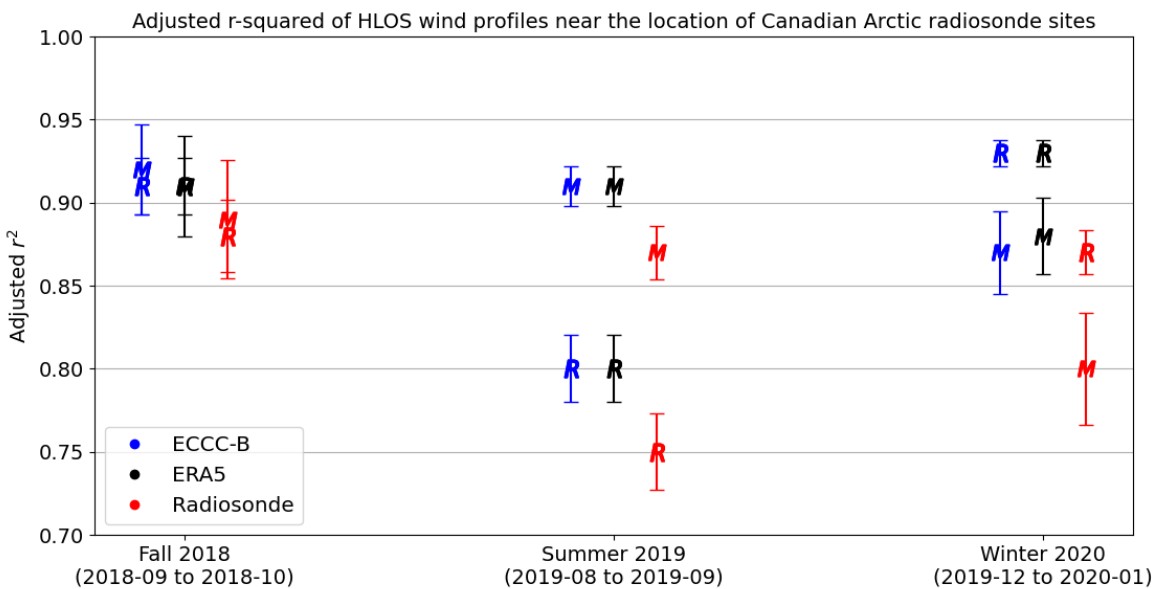

**Figure 6.** Adjusted r-squared of vertical HLOS wind profiles with 99% confidence level from coincident Aeolus (Rayleigh, R, and Mie, M)
overpasses near radiosonde stations over the Canadian Arctic (shown in Figure 1) and ECCC-B, ERA5, and radiosonde measurements during
fall 2018 (early FM-A), summer 2019 (early FM-B), and winter 2020 (mid-FM-B).

A systematic difference between the three measurement periods is apparent. Rayleigh winds could be very sensitive
to the solar background radiation (SBR) that contaminates the weak Rayleigh backscatter signal under clear sky condition.
Random errors caused by the SBR were anticipated. Aeolus points towards the sun-synchronous night-side of its orbit to
minimize the impact of SBR on the wind observations (Kanitz et al., 2019; Zhang et al., 2020). However, the impact is greater
than expected, especially during summertime over the Arctic where the Rayleigh random errors can be as high as 8 ms$^{-1}$ (Zhang
et al., 2019; Krisch et al., 2020, Reitebuch et al., 2020). As a result, as will also be shown below, the consistency of Aeolus
Rayleigh winds with other datasets markedly worsens during summer. We also note a slight drop in consistency of the Mie
winds for the mid-FM-B period, which took place in winter 2020: for instance, the adjusted r-squared and their 99% confidence
intervals, between Mie winds and ECCC-B, are 0.92±0.03 during fall 2018, 0.91±0.01 during summer 2019, and 0.87±0.02
during winter 2020. The cloud cover, number of observations, and estimated error from Aeolus do not seem to control this
decrease. Its cause, which could be due to the Aeolus measurement or the wind retrieval, remains unclear.

### 3.2 Pan-Arctic validation against background and reanalysis products

For more insight into how the behavior in the Canadian datasets we have examined extends to other Arctic regions, we now
evaluate Aeolus wind measurements over the whole Arctic (measurement and profile counts provided in Table S3), including
over the Arctic Ocean, where wind observations are particularly sparse. We evaluate the HLOS winds in relation to the ECCC-

B and ERA5 products poleward of 70° N. Note that we exclude the measurements over a region that partially covers Greenland, North Atlantic Ocean, and Iceland (50° W to 5° E and 52.5° N to 80° N) in September 2019 because Aeolus had a different

range bin setting over this area for AVARTAR-I campaign purposes (Fehr et al., 2020). The time-series of the estimated errors from 2B06 (solid line) and 2B10 (dashed line) datasets and the root-mean-square difference (RMSD) between the Aeolus Rayleigh winds and ECCC-B data are shown in Fig. 7. The estimated errors and RMSD over the excluded region (blue) have a sudden jump on 9 September indicated by the vertical red dashed line while the rest of the Arctic (black) shows a consistent decrease in estimated errors and RMSD. This decrease in this period also shows how the contribution to the error due to the

solar background radiation is decreasing with the transition from summer to fall conditions. The reprocessed data has improved estimated errors and RMSD over the excluded region; however, the jump is still visible. From 9 September to 6 October 2019, the satellite was measuring at a finer vertical resolution to compare with research-flight measurements. Thus, the derived winds were averaged over fewer measurements. The Rayleigh winds are particularly noisy due to the loss in optical signal along the atmospheric and internal path (Reitebuch et al., 2020), which emphasizes the seasonal variation of the solar background noise

during boreal summer and perhaps also reflects the attempt to measure finer vertical scales. Thus, the price for having a higher vertical resolution is larger errors for this specific range bin setting. For the consistency of the data quality, we thus exclude the measurements for this period and region from subsequent analysis.

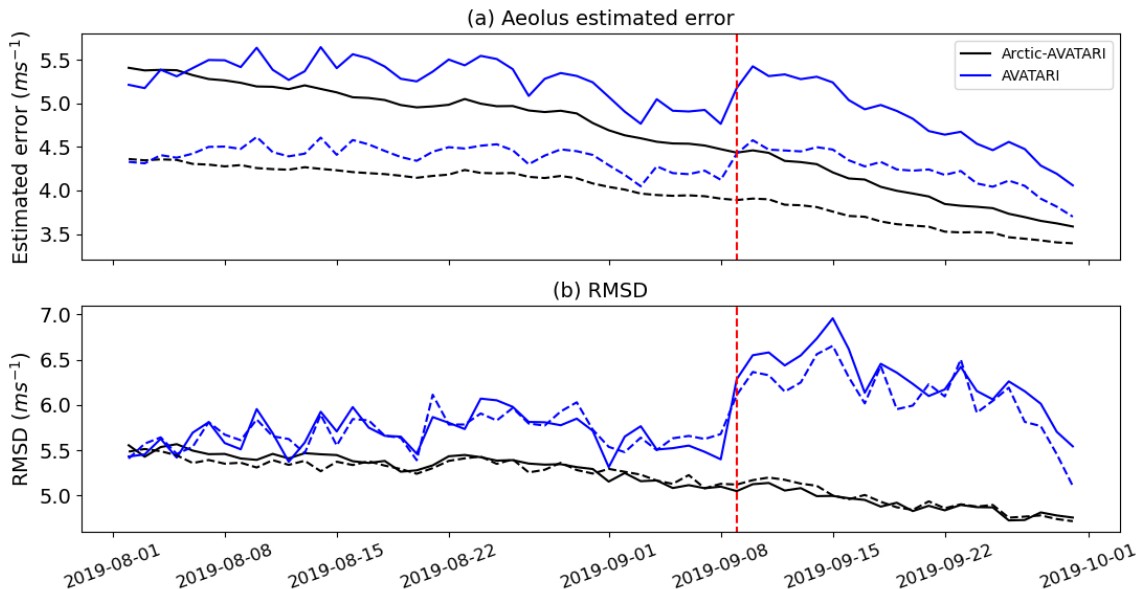

**Figure 7.** Time-series of (a) Aeolus L2B estimated error of Rayleigh winds and (b) RMSD between the ECCC-B and Aeolus 2B06 (solid)
and 2B10 (dashed) data from 2 August to 30 September 2019. The data is averaged over the region with a different range bin setting for the AVATAR-I campaign (blue) and over the rest of the Arctic (black).

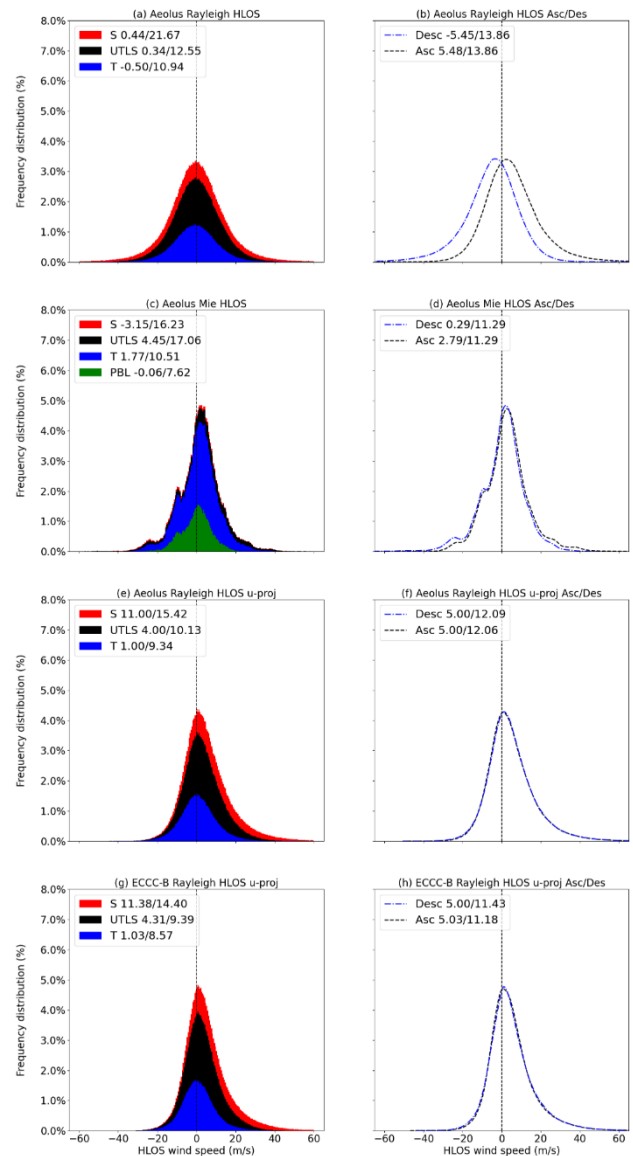

**Figure 8.** Aeolus Rayleigh ((a) and (b)) and Mie ((c) and (d)) HLOS measurement stacked frequency distributions (%) during winter 2020 over the Arctic (>70° N). The HLOS winds are projected onto the east-west directions ((e) to (h)) from Aeolus Rayleigh measurements and ECCC-B HLOS winds. The panels on the right show the distribution of ascending and descending measurements separately. The means and standard deviations of distributions in each atmospheric layer are listed in the figure legends.

By expanding the region of analysis, we obtain a larger sample, which allows us to look at the separate ascending and descending orbit phases, frequency distributions in different layers in the atmosphere, correlations along the Aeolus track in different atmospheric layers, and the geographic variation of correlations between vertical HLOS profiles. We define four

atmospheric layers: the planetary boundary layer (PBL, in the vertical range up to 2 km), the free troposphere (T, 2-8 km), the upper troposphere/lower stratosphere (UTLS, 8-16 km), and the stratosphere (S, altitudes greater than 16 km). Figure 8a and c show examples of stacked distributions of the Rayleigh and Mie winds for winter 2020 over the Arctic. Rayleigh winds at pressure greater than 850 hPa are ignored as recommended by Rennie and Isaksen (2020), because they show some indications of degradation in forecasts. The Mie channel measures winds under cloudy conditions and thus has more measurements in the PBL and in the free troposphere than at higher altitudes (e.g., Fig. 8c). Furthermore, the ascending and descending Rayleigh distributions (Fig. 8b) are symmetric about zero due to the symmetric azimuth angle of the instrument with respect to the north when switching from the ascending to the descending phase. To avoid this artefact and to add some insight into the wind features being measured, we also compare the projected HLOS wind vector into its zonal (positive to the east) and meridional (positive to the north) components. The stacked distribution of the zonal-component of the HLOS winds is shown in Fig. 8e and g for Aeolus Rayleigh and ECCC-B HLOS winds. By doing this projection, the distributions for ascending and descending measurements are brought into better agreement (Fig. 8f). We also notice that the projected zonal component of the HLOS winds can provide some information about the vertical variation of the zonal wind. For example, for Aeolus Rayleigh, the mean values of the zonal projection of the HLOS wind for the stratosphere, UTLS and troposphere are 11.00 ms$^{-1}$, 4.00 ms$^{-1}$ and 1.00 ms$^{-1}$ respectively. These mean values, as well as their standard deviations (see legend of Figs. 8e and g), agree well with ECCC-B (and ERA5 – not shown). Aeolus-measured positive values of the zonal wind component from the stratosphere into the troposphere is consistent with the known climatological presence of westerlies in this region in polar winter. Analyzing the zonal projection of the HLOS winds highlights this feature.

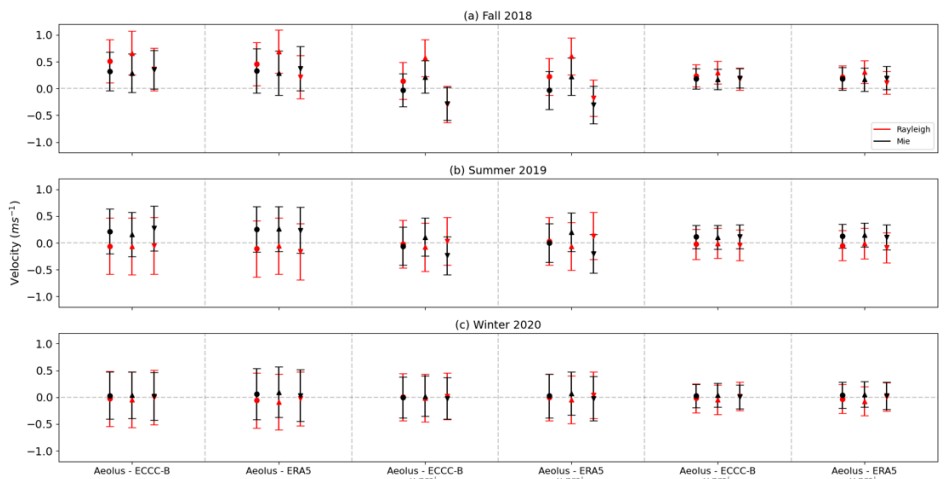

**Figure 9.** Means ($10^0$) and standard deviations ($10^1$) of the distributions of the differences between Aeolus Rayleigh (Mie) winds and ECCC-B and ERA5 shown as red (black) dots or triangles, and error bars, over the Arctic during (a) boreal fall 2018, (b) summer 2019, and (c) winter 2019-20. The dots represent Aeolus measurements in the atmosphere which can be decomposed into ascending (upright triangles) and descending (inverted triangles) measurements. The distributions of the differences of the projected winds are shown in the latter four columns.

We compare the distributions of the differences between the Aeolus wind measurement data and the ECCC-B and
ERA5 data during fall 2018, summer 2019, and winter 2020 over the Arctic, as summarized in Fig. 9, which shows the bias
and standard deviations of the differences for the HLOS winds and for their zonal and meridional projections. To highlight the
variations of the means, the standard deviations were divided by a factor of 10. The measurements are separated into Rayleigh
(red dots) and Mie winds (black dots). They are further separated into ascending (indicated with upright triangles) and
descending (inverted triangles) measurements. The results, with the biases being smaller than 0.7 ms$^{-1}$, are consistent with
ECCC bias correction method. The means and standard deviations of the differences in the ascending and descending
measurements do not show a significant difference. The discrepancies in the meridional projections of the HLOS winds are
smaller because Aeolus picks up mostly the zonal component of the winds due to the direction of the LOS.

Although Fig. 9 shows an overall agreement between Aeolus, ECCC-B, and ERA5, more analysis is required to bring
out the differences between the datasets. One way to do so is to separately investigate the consistency between Aeolus and
ECCC-B or ERA5 HLOS winds in the PBL, troposphere, UTLS, and stratosphere. Figure 10 shows normalized Taylor
diagrams (Taylor, 2001), with ECCC-B as reference, for Aeolus Rayleigh and Mie measurements over the Arctic during the
three seasons of analysis. The angle indicates the correlation between Aeolus measurements and ECCC-B. The distance to the
origin represents the standard deviation and the distance to the star (reference point (1,0)) represents the RMSD; both statistics
are normalized by the standard deviation of reference data. The 1.0 normalized standard deviation is highlighted; data that falls
outside the dashed quarter circle is noisier than the reference data. Figure 9 shows that Aeolus data consistently has greater
standard deviations than ECCC-B during all three periods and for both Rayleigh and Mie winds: its normalized standard
deviations are typically within 1.05 to 1.40 or greater. This might imply that Aeolus provides noisier data, that the ECCC-B is
missing some extreme values in its wind-component distribution, or both. However, the RMSD are generally within one
normalized standard deviation and correlations are normally greater than 0.8 for Rayleigh winds above the troposphere and
for Mie winds below the lower stratosphere, with an exception for stratospheric consistency for Rayleigh winds during the
summer. During the boreal summer period, the data in the stratosphere seem to agree less with the ECCC-B data, reflecting
reduced sampling, solar background noise that is most effective during summer, changes in the atmospheric environment in
summer, thermal changes in the telescope, and other possible errors (Reitebuch et al., 2020).
Generally, the Rayleigh-clear channel provides consistent data with the ECCC-B through the troposphere (T in Fig.
10), the UTLS (U) and the stratosphere (S), while the Mie-cloudy channel provides consistency from the PBL (B) to the lower
stratosphere. This reflects the vertical sampling and instrument characteristics and reveals effective complementarity of the
instrument and retrieval design. For this reason, in the next paragraph, where we investigate the spatial distribution of the
consistency in the lower and upper atmospheric regions, we exclude the Rayleigh winds in the PBL and the Mie winds in the
stratosphere.

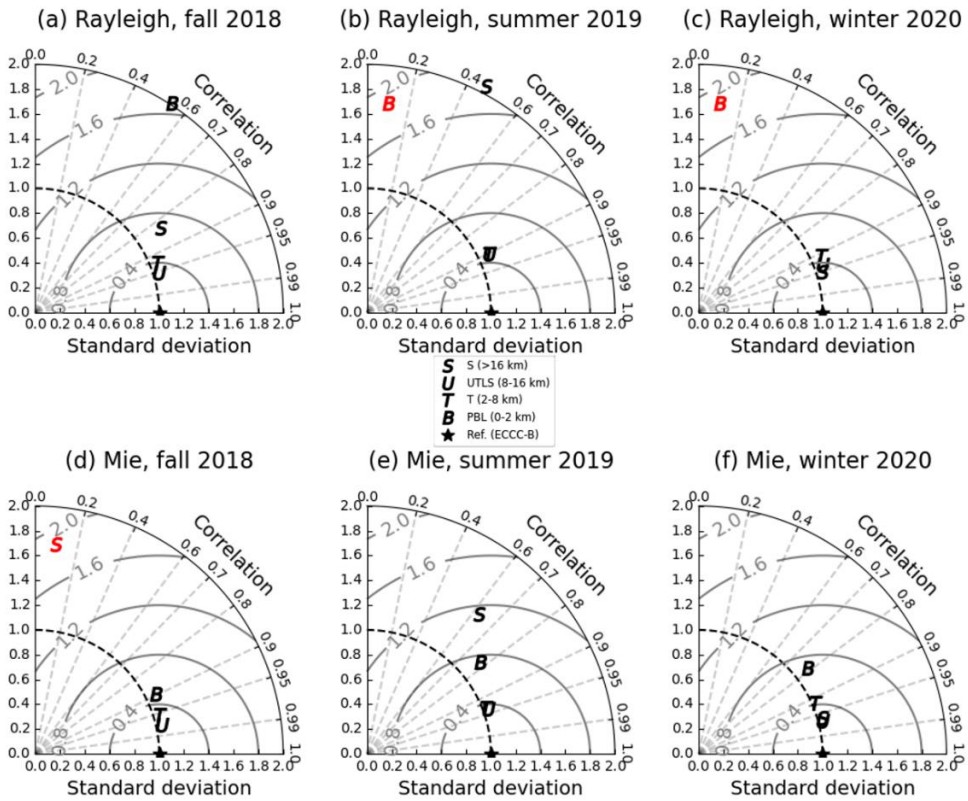

**Figure 10.** Normalized Taylor diagrams with ECCC-B as references for Aeolus measurements over the Arctic. The correlation coefficients and standard deviations are calculated for each layer. The angle indicates the correlation between Aeolus measurements and the reference. The distance to the origin represents the normalized standard deviation and to the star (reference) represents the normalized root-mean square error. Rows are distinguished by channels; columns are distinguished by seasons. The red markers on top left of panels represent layers with normalized standard deviations that are outside the range shown (> 2.2).

Figure 11 shows the RMSD between Aeolus and ECCC-B for Rayleigh tropospheric (T) and Mie PBL + tropospheric (B+T) profiles, and Fig. 12 shows the same but for Rayleigh UTLS + stratosphere (U+S) and Mie UTLS (U) measurements. Since the estimated errors and RMSD were consistently decreasing in September 2019 over the Arctic except for the region with different range bin setting (Fig. 7), to avoid misinterpretation of the results on the maps, we exclude the data during this period over the entire Arctic. Since the measurement density differs depending on the latitude, the RMSD of the profiles are calculated over nearly equal surface area, using the Equal-Area Scalable Earth (EASE) Grids (Brodzik et al., 2012). Each grid cell is around $10^4$ km$^2$ which is approximately the square of the along-path resolution of Aeolus Rayleigh winds. The first two and the last columns represent the distributions using the near real-time 2B02/06/07 datasets; the third column shows the distributions using the reprocessed 2B10 data during the early FM-B period.

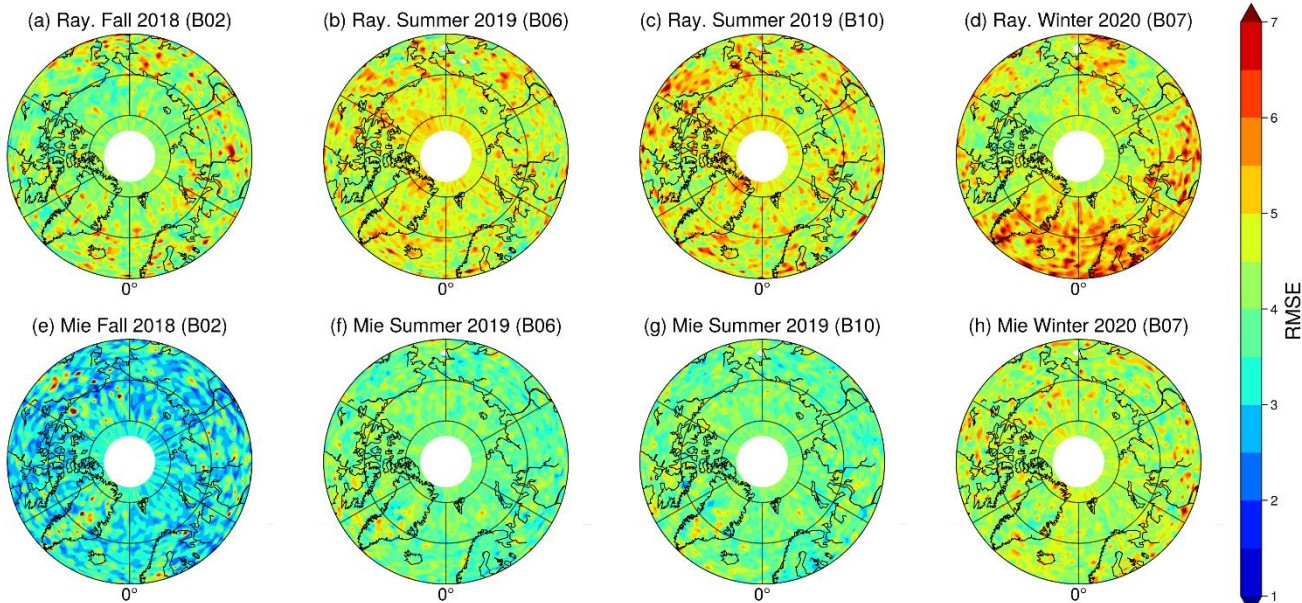

Figure 11. RMSD of Aeolus and ECCC-B vertical HLOS wind profiles for selected lower-atmospheric regions (Rayleigh T and Mie B+T) during fall 2018, summer 2019, and winter 2020.

The distributions of the RMSD in the lower atmosphere are relatively homogeneous across oceanic, ice-covered, and continental regions. This suggests that the overall good agreement seen for the in-situ Iqaluit and Whitehorse data as well as the northern Canadian region in the radiosonde network extends from land to the ocean regions without obvious systematic differences in consistency. The agreement between Aeolus and ECCC-B for the Mie winds is better than for the Rayleigh winds for all three periods of analysis. The RMSD for the Mie winds largely lie between 2 ms$^{-1}$ and 4 ms$^{-1}$, and for Rayleigh winds, the RMSD are generally greater than 4 ms$^{-1}$. This was anticipated because the Rayleigh winds are noisier for reasons alluded to above. The RMSD are systematically greater in the upper-atmosphere than in the lower-atmosphere as shown in Figs. 11 and 12, and the differences could be anticipated from the Aeolus estimated errors from L2B product (as shown in Figs. S1 and S2).

In the upper atmosphere, the RMSD are significantly larger for the Rayleigh winds during FM-B period, especially over Greenland where the RMSD are greater than 6.0 ms$^{-1}$. This might be caused by the reflection from the Greenland ice sheet which leads to enhanced errors. The estimated errors also show a similar pattern: the estimated error over Greenland in summer 2019 is around 6.0 ms$^{-1}$ (Fig. S2b), the reprocessed product reduced the estimated error to around 5.0 ms$^{-1}$ (Fig. S2c), and it is around 4.0 ms$^{-1}$ during winter 2020 (Fig. S2d) due to the lack of solar background noise in the boreal winter. This pattern is however not seen during the FM-A period. But whatever the source of these changes (e.g., summertime solar background noise amplification or calibration errors coinciding with the start of laser-B stream), it is noteworthy that the

estimated error product contains potentially useful information for validation purposes. Because such information is useful for error characterization in NWP, as we will discuss below, a more detailed investigation into the estimated error product, including its seasonal, geographic, and flow dependence, is warranted.

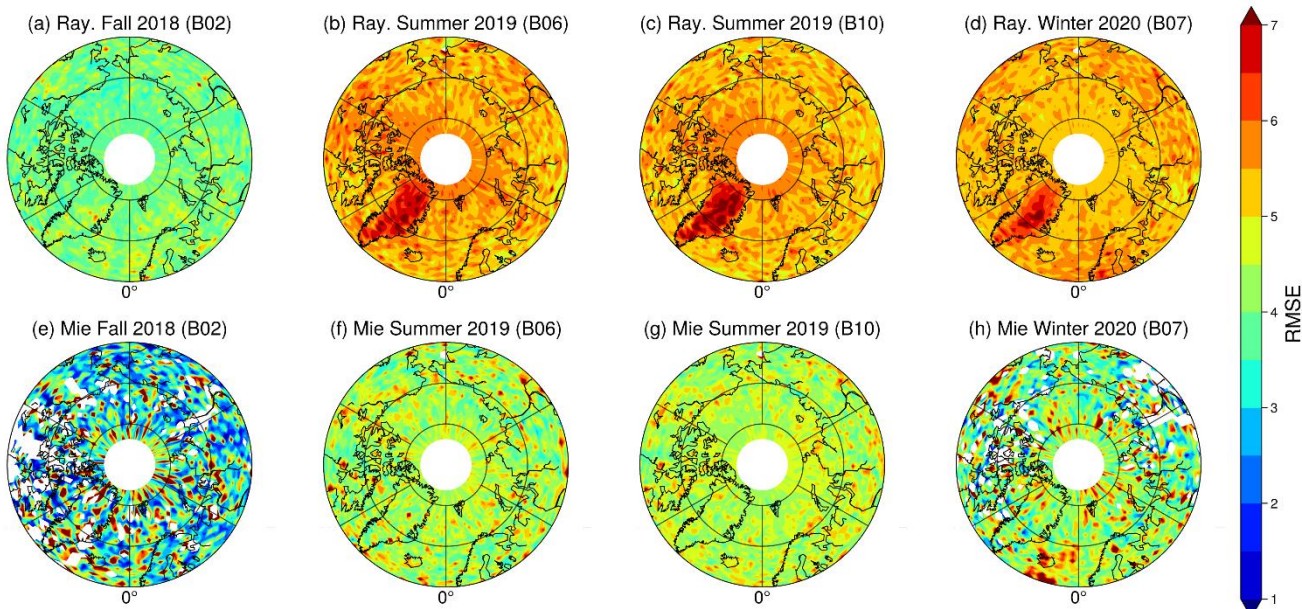

**Figure 12.** Similar to Fig. 11, but for selected upper-atmospheric regions (Rayleigh U+S and Mie U).

Note that the first reprocessed data, 2B10, only overlaps with one of the three periods of study: August to September 2019. The estimated observational errors have decreased compared to the 2B06 data (Figs. S1 and S2) since the bias due to the M1 mirror temperature dependence is updated on a daily basis and the dark current signals have been removed using 480 improved quality control. However, we do not see the same improvement in the O-B statistics between 2B06 and 2B10 products over the Arctic region, possibly because the reprocessed product had improved the precision (error characterization) of the measurements while not leading to a change in the overall agreement between the products, suggesting that accuracy is not changed.

## 4 Summary and conclusion

In August 2018, ESA successfully launched the first spaceborne DWL Aeolus to measure global wind profile measurements along its LOS, using the instrument's Rayleigh and Mie channel receivers. Only Rayleigh-clear winds and Mie-cloudy winds are considered in the validation. In this work, the Aeolus data product are bias-corrected and quality controlled using the quality flag from the L2B product, estimated error screening following ECMWF's guidance, and the screening when O-B is greater than 30 ms$^{-1}$ to remove any additional outliers. Our results show consistent Aeolus data products around the sites with

the 9-h short-range forecast from ECCC ("background", ECCC-B), reanalysis ERA5 from ECMWF, and in-situ measurements using radiosondes and Ka-band radar, for the period 15 September to 16 October 2018. For example, the adjusted r-squared between Aeolus Rayleigh winds and ECCC-B is 0.92, and 0.91 between Aeolus and ERA5 at the Iqaluit site. For the Aeolus Mie winds, the statistical results are 0.87 with ECCC-B and 0.81 with ERA5.

The comparison with the Ka-band radar at Iqaluit has been limited to the early phase of Aeolus lifetime due to a technical failure of the ground-based radar requiring extensive repair. The agreement between Aeolus wind product and the Ka-band radar is systematically worse than with the forecasts and reanalysis products. Nevertheless, for the limited sampling available, we have verified that the Ka-band radar provides consistent vector winds above 200 m a.g.l. with a bias less than 1 ms$^{-1}$ and a RMSD less than 3 ms$^{-1}$, and that the radiosonde measurements are consistent with Aeolus winds with an adjusted

r-squared greater than 0.8. While the Ka-band radar data availability was limited, its analysis is retained in this paper because this independently generated ground-based data, which is unique in the large geographical region of the Canadian North, is not assimilated in any NWP system and is critical to validate Aeolus in this part of the Arctic. This highlights that radar observations are rare and challenging to obtain because of costs and logistics and the sparsity of independently generated ground-based data in the Arctic. Because this critically limits validation capacity for this region, these results encourage

programs like CAWS to enhance independent radar measurements over the Canadian Arctic and to continue investment in such infrastructure.

We also validate Aeolus wind products with ECCC-B, ERA5, and radiosonde measurements around other radiosonde sites for the periods 15 September to 16 October 2018, 2 August to 30 September 2019, and 1 December 2019 to 31 January

2020. This comparison raises the issue of solar background noise at high latitudes during summertime, which degrades the adjusted r-squared of the Rayleigh winds by about 10% during the early FM-B period (Fig. 6). This issue extends to the analysis of the pan-Arctic, where the effect of solar background radiation is even larger over polar regions where there are 24 hours periods of sunlight in summer (Figs. 12b-c).

In our analysis of the pan-Arctic region, we found an overall agreement by comparing the distributions of the HLOS winds, ascending and descending HLOS winds, and projections of HLOS winds onto east-west and north-south directions in different atmospheric layers (Fig. 8), and we also compared the distributions of the differences between Aeolus and ECCC-B and ERA5 (Fig. 9). Due to the angle of the HLOS, when comparing the distributions, separating the ascending with descending measurements helps avoid cancelling out part of the HLOS winds and projecting the HLOS winds on to zonal and meridional

directions provides some insight on the vertical variation of the HLOS winds. To further investigate the consistency, we showed normalized Taylor diagrams in Fig. 10, which reveal that the standard deviations of Aeolus winds are 5 to 40% greater than ECCC-B in every layer with abundant Aeolus measurements, i.e., above the troposphere for Rayleigh winds and below the lower stratosphere for Mie winds, with an exception for the stratospheric Rayleigh winds in summer. Future work could

investigate whether this discrepancy arises because Aeolus provides noisier measurements due to limitations of the processed observations or because Aeolus is measuring structural detail not captured in the forecast and reanalysis. In any case, this analysis reveals consistent Rayleigh HLOS winds above the troposphere and Mie HLOS winds below the lower stratosphere with correlations higher than 0.8 except during summer in the stratosphere due to the solar background noise and normalized standard errors within one standard deviation of ECCC-B. Finally, the spatial RMSD of Aeolus and ECCC-B vertical wind profiles confirm their mutual consistency. We found that the spatial variability of the time-averaged L2B estimated error product is in good agreement with the spatial variability of the RMSD between Aeolus and ECCC-B HLOS winds over the Arctic region. This validates the use of L2B estimated error product as a predictor for the HLOS wind observation errors in data assimilation systems, as proposed by Rennie et al. (2021), to obtain optimal positive impacts on forecasts from assimilating Aeolus winds.

In conclusion, the mutual consistency between Aeolus and the short-range forecast and reanalysis over the Arctic suggests that Aeolus provides reliable wind measurements that can further advance our knowledge on circulation and further improve current NWP models, and also suggests that the ERA5 reanalysis and ECCC-B provide good estimates of the circulation over the Arctic, reflecting the volume of satellite data assimilated daily (over 1 million observations, mainly radiances) and mass balance constraints that hold at high latitudes. It remains open, however, how the consistency between Aeolus and available analyzed data depends on horizontal and vertical scale. It is reassuring that this consistency is seen in a vertical range extending from the planetary boundary layer in the Mie channel in all three observation periods to the stratosphere in the Rayleigh channel (in the non-summer periods). The promise of added value to forecasts from Aeolus winds is already being borne out at several centers that have assimilated Aeolus data into their operational NWP model and are seeing positive impact (e.g., Rennie et al., 2021). A focus on predictability of weather systems in the Arctic, and on predictability from wind information centred in the Arctic, is the subject of current work.

**Data availability**

The Ka-band radar at Iqaluit and ECCC-B data used in this paper can be provided by the corresponding author (gina.chou@mail.utoronto.ca) upon request. The ERA5 data can be downloaded from the Copernicus Climate Change Service (C3S) Climate Date Store (https://cds.climate.copernicus.eu/cdsapp#!/dataset/reanalysis-era5-pressure-levels?tab=overview). The radiosondes data can be downloaded from http://weather.uwyo.edu/upperair/sounding.html. Aeolus L2B data can be obtained from the VirES visualization tool (https://aeolus.services/).

**Author contribution**

Chih Chun Chou prepared the main part of the paper and performed the statistical analyses of Aeolus data, in-situ measurements, ECCC-B, and ERA5 data. Paul Kushner and Stéphane Laroche guided Chih Chun Chou for the analyses and the written part. Stéphane Laroche provided the ECCC-B data and Zen Mariani provided the in-situ measurements at Iqaluit

and Whitehorse supersites. Zen Mariani, Peter Rodriguez, and Stella Melo provided guidance in reading and retrieving the radar data. All co-authors helped review the manuscript.


**Competing interests**

The authors declare that they have no conflict of interest.

**Acknowledgements**

We acknowledge ESA for the Aeolus data, ECMWF for the ERA5 data, and ECCC for the short-range forecast and in-situ measurements. We acknowledge the support of the Canadian Space Agency's Earth System Science Data Analysis program.

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
