# Peer review of "Validation of the Aeolus Level-2B wind product over Northern Canada and the Arctic"

_Atmospheric Measurement Techniques, 2021_

## Referee Comment (RC2)

**General comments:**

The authors compared the Aeolus wind measurements with Ka-radar, wireless soundings, ECCC and ERA5 reanalysis data, and the results show that the Aeolus wind measurements are in good agreement with other wind measurements or reanalysis data. This paper has important implications for the application of Aeolus wind measurements in the Arctic, where wind measurements are currently scarce. However, the paper requires significant revision before publication, and the specific issues are described below.

1. The authors use other soundings and reanalysis data as a benchmark for comparison with the Aeolus wind data, but the data quality of the other data is not presented in the paper. Perhaps the other wind data also have large biases in the Arctic, and then the authors' use of them as a comparison benchmark would make none of the comparisons in this paper credible. Especially for high altitude wind fields, the data sets may have different data quality performance at different heights. Moreover, the reanalysis dataset itself contains assimilation of existing sounding data. The authors need to fully justify the reliability of the data in each of the datasets used in this paper.

2. The authors' presentation of the spatio-temporal matching process between the datasets is not clear enough. It is suggested that the data matching process be introduced as a separate section and the information about the data used in this paper be summarized. A clear and detailed data matching process is desired to be seen.

3. the authors' analysis of Figure 5 is not rigorous enough and the conclusions drawn from Figure 5 are not reliable enough, see Specific comments L296.

4. The authors need to clarify the practical significance of the discussion of the statistical distribution of the wind products themselves in Figures 7 and 8. It might be more valuable to discuss instead the distribution of the difference between the Aeolus data and the other data.

**Specific comments:**

L82:*1 December to 31 January 2020*
Wrong time markings

*L115: ECMWF has recently published the first reprocessed data (2B10)*
Data links should be added after the data introduction.

*L125: The quality control recommendation following the Guidance for Aeolus NWP Impact Experiments (Rennie and Isaksen, 2019), including the threshold for L2B estimated observation errors.*
The recommendation given by NWP is a threshold range. Authors should submit specific thresholds to be used when processing data.

*L129: We further reject the outliers by excluding all the data when the difference between the observations and ECCC-B or ERA5 is greater than 30 m/s.*
For the screening threshold of 30m/s, the authors need to give an explanation, either from data

analysis or literature.

*L162: ECCC-B is then linearly interpolated to Aeolus measurement locations and times.*
The process of linear interpolation needs to be clarified. In addition, the main comparison data in the latter section does not provide complete information on the vertical resolution, and due to the large number of datasets used in this paper, it is recommended to use a table after section 2.5 to summarize the important information of each dataset used.

*L165: Reanalysis ERA5*
For the ERA5 dataset, assimilation relationships between it and the other datasets used in this paper should be added.

L190-L210:
From the latter, there is very little overlap between Ka-band radar or LIDAR with Aeolus data. The main object of comparison is the radiosondes, so its data should have been presented in more detail. In particular, the horizontal drift problem of radiosondes, which seems to be not taken into account by the authors, may also lead to a large bias. In addition, is the introduction and use of Ka-band radar and ground-based LIDAR data in this paper meaningful? It seems that the absence of these two does not affect the logic and conclusions of this paper.

*L246:On 24 September, Aeolus measures westerly winds in reasonable overall agreement with the other data.*
The agreement between Aeolus and the other data in Figure 2b does not seem to be obvious, and the deviation in some data points is already close to 50%.

L249:
Figures 3 and 4 are identical and need to be modified. The comparison of Ka-band radar in Figure 3d needs to be addressed for its significance, since there are only about 10 data points. Also the information on the fitted straight lines, standard deviation and sample size in the figure can be added, while the information in the supplemental Figure S1 will be placed in the original text in the form of a table.

L270:
The data consistency performance of the two sites is different, and the data consistency of ERA5 and ECCC-B with Aeolus in the Whitehorse site didn't change, and the conclusion given by the authors as well as the explanation is not reasonable enough.

L275-L292:
The authors' discussion of Ka-band radar consistency with Aeolus and its causes seems unnecessary for this paper, as there are too few overlapping data points and no valid conclusions are drawn.

L296:

Are the sample sizes in Figure 5 the same for the three time periods? The use of the expression summer, fall, and winter is not rigorous and should be specific to dates. In addition, the discussion of Figure 5 is inadequate. If solar radiation is used to explain Figure 5, then why did the overall performance of fall 2018 be better than that of winter 2020? The Mie channel also performed better in summer 2019 than in winter 2020, and the Mie channel is also influenced by solar background radiation. The authors seem to have overlooked some phenomena in their haste to reach conclusions.

*L355:Since ECCC-B and ERA5 are mutually consistent*

This may need to be more fully demonstrated.

L359-L364:
The conclusion of this paragraph does not need to be obtained by data analysis. It is just a mathematical law. The discussion of the longitudinal and latitudinal components also seems to be unnecessary.

Figure 8:
What is the significance of discussing the mean and standard deviation of the wind speed samples themselves? Clarification by the authors is needed. It may be more meaningful to discuss the distribution of the differences between the Aeolus wind measurement data and the ECCC-B and ERA5 data.

*L377: Figure 8 shows an overall agreement between Aeolus, ECCC-B, and ERA5*
The proof of consistency between data by comparing data distribution characteristics only is not enough.

*L391: Figure 9 shows that Aeolus data consistently has more structure than ECCC-B during all three periods and for both Rayleigh and Mie winds.*
What does "more structure" mean here? Please explain.

*L395: During the boreal summer period, the data in the stratosphere seem to agree less with the ECCC-B data, reflecting reduced sampling, solar background noise that is most effective during summer as mentioned earlier, and other possible errors (Reitebuch et al., 2020).*
The derivation of this conclusion is not rigorous, there are many possible reasons for the decrease in the correlation between the Aeolus data and ECCC-B data in the stratosphere in summer, and it is also possible that it is caused by changes in the atmospheric environment in summer, thermal changes in the telescope, etc. I don't think we can make a speculation on the cause from Figure 9. But the authors seem to attribute it to solar radiation in the abstract.

*L402: For this reason, in the next paragraph, where we investigate the spatial distribution of the consistency in the lower and upper atmospheric regions, we exclude the Rayleigh winds in the PBL and the Mie winds in the stratosphere.*
It would have been more convincing if the authors had made this data trade-off from the analysis

of data used in this paper. Although Aeolus was designed to complement the dual channels at altitude, there are many cases where the Mie channel has higher data volume and data quality at altitude than at lower altitudes, especially in summer. Simply removing the data would be detrimental to the subsequent analysis.

Figure10, Figure11:
The calculation process of RMSD in this paper should be clearer, and it is better to give the formula and the range of RMSD to be considered "consistent". In addition, the radial mutations in these two figures seem to be inconsistent with common sense, especially in Figure 11e, which I hope the authors can explain. Also, the same color scale should be used for all subplots. The number of data samples used for different subplots should be provided, because the valid sample size may vary greatly for different seasons at different altitudes. In addition, the data density of Aeolus is also different for different latitudes, how did the authors deal with this point, and does this lead to lower quality data for lower latitudes?

*L446: No significant improvement is seen here because we have implemented a weekly updated dynamic bias correction to the near real time data.*
How do the authors explain the slightly higher RMSD of 2B10 data compared to 2B06 in Figure 10 and Figure 11? What is the meaning of the dynamic bias correction mentioned in the paper, please elaborate, and what is the difference between this correction and the reprocessed data correction in the L2B product?

*L476: We have found some initial evidence that the estimated error product is also a good predictor of RMSD between Aeolus and the reanalysis, which could be useful for constraining future forecasts.*
The "estimated error product" itself is used to estimate the difference between the Aeolus wind product and the true wind field. If it does not predict the RMSD between Aeolus and reanalysis data, then the reanalysis data deviates from the real wind field. I don't understand the purpose of the author's statement, perhaps more information about "constraining future forecasts" is needed.

---

## Author Comment (AC1)

Referee comment on "Validation of the Aeolus Level-2B wind product over Northern Canada and the Arctic" by Chih-Chun Chou et al., Atmos. Meas. Tech. Discuss., https://doi.org/10.5194/amt-2021-247-RC1, 2021

We thank the reviewers for their careful reading of the manuscript and comments, which helped us improve the study.

**General comments:**

The authors compare Aeolus winds with Ka-radar and radiosonde winds in Northern Canada in periods of the early phase during the first laser nominal flight model (FM-A; 2018-09 to 2018-10), the early phase during the second flight laser (FM-B; 2019-08 to 2019-09), and the mid-FM-B periods (2019-12 to 2020-01). They also compare Aeolus wind fields with ECCC-background and ERA5 reanalysis wind fields over the whole Arctic (poleward of 70ºN). Since direct wind observations are especially sparse in the Arctic, the topic is interesting and has important implication in evaluating the quality of Aeolus winds over the Arctic. However, the following major issues need to be improved.

1. In order to use Aeolus wind data in numerical weather prediction models and to improve the data quality in newer processor versions, the systematic and random errors of Aeolus winds must be understood. As the purpose of this study is to evaluate the quality of Aeolus wind products over Northern Canada and the Arctic in comparison with several available observational products, the systematic and random errors of Rayleigh-clear and Mie-cloudy winds over these regions should be evaluated. Although the authors evaluated the random errors using Figs. 9–11, they did not evaluate the systematic errors.

   Thank you for pointing out the need to clarify this aspect. Systematic errors were not fully examined in this work because bias-corrected Aeolus data were used in this study according to guidance provided by Rennie and Isaksen (2020). As stated in the paper, a bias offset of -1.35 ms$^{-1}$ was added to the FM-A 2B02 Mie-cloudy winds, and a look-up table bias correction scheme was applied at ECCC to the FM-B 2B06/07 Mie-cloudy and Rayleigh-clear winds, as described in Rennie and Isaksen (2020). The FM-B 2B10 were bias corrected in the L2B processor, based on the M1 mirror temperature variations.

   The following sentence is added in Line 62 for clarification: "We will focus on analyzing random errors instead of systematic errors since, as recommended for operational NWP practice, bias corrected Aeolus data is used in this study (see Sect. 2.1)."

Furthermore, instead of representing Aeolus, ECCC-B and ERA5 winds separately, Fig. 8 now shows the means and standard deviations of the *differences* between Aeolus and ECCC-B and ERA5. The means of the differences therefore reflect the remaining bias between the datasets after the dynamic bias correction has been applied. The associated paragraph describing Fig. 8 is also revised. Starting at line 394:

"We compare the distributions of the differences between the Aeolus wind measurement data and the ECCC-B and ERA5 data during fall 2018, summer 2019, and winter 2020 over the Arctic, as summarized in Fig. 8, which shows the bias and standard deviations of the differences between Aeolus HLOS winds and the ECCC-B HLOS winds, and ERA5 HLOS winds, and their zonal and meridional projections. The measurements are decomposed into Rayleigh (red) and Mie winds (black). They are further decomposed into ascending (indicated with upright triangles) and descending (inverted triangles) measurements. The results, with the bias (the mean values of these differences for the different sampling used) being smaller than 0.7 ms$^{-1}$, are consistent with our bias correction method. The distributions of the differences in the ascending and descending measurements do not show a significant difference. The discrepancies in the meridional projections of the HLOS winds are smaller because Aeolus picks up mostly the zonal component of the winds due to the direction of the LOS."

2. My recommendation concerns bringing the findings of the paper into perspective with what is known from other literature, e.g., Belova et al. (2021), which conducts the validation of Aeolus HLOS winds against ground-based radar measurements in the Antarctica and northern Sweden.

Belova et al.'s (2021) findings on the systematic and random errors are summarized in line 72: "In related Arctic-based work, Belova et al. (2021) have found consistency between Aeolus winds and a ground-based radar situated in northern Sweden with insignificant biases between the two products (less than 1 ms$^{-1}$) and slightly increased random errors for Aeolus in the boreal summer, possibly due to sunlight scatter."

3. During the mid-FM-B period (1 December 2019 to 31 January 2020), Aeolus L2B near real-time baseline products '2B07' were used. Please check https://aeolus-ds.eo.esa.int/oads/access/collection/L1B_L2_Products/tree.

This is fixed now. Thank you for pointing this out.

4. The total number of measurements (N) and number of profiles (p) are important in calculating adjusted r-squared in Figs. 3, 4, and 5. Please give N and p of each site and period.

   Thank you. The number of measurements (N) and number of profiles (p) are now provided in Table S1 to S3. Table S1 is for the validation at the sites, Table S2 for validation over the Canadian Arctic, and Table S3 for validation over the pan-Arctic.

5. Figure 3d shows the scatter plot between Aeolus Rayleigh wind and Ka-band radar at Iqaluit. The number of comparison pairs is only 11. In my opinion, the sample size is too small. I suggest that the authors must perform the significance test.

   Yes, we acknowledge that there are few applicable samples here. In order to assess significance, an F-test is performed, and all comparisons are at 99% confidence level, including the comparison between Aeolus and Ka-band radar at Iqaluit. We try to acknowledge more clearly the situation at line 311:

   "Generally, the sampling for these radar measurements is highly limited, which tends to reduce the agreement compared to the other datasets. Nevertheless, the agreement on the variances between Aeolus and the Ka-band radar is at 99% confidence level using F-test. This analysis highlights the importance for programs such as CAWS to continue to provide ground-based radar measurements to ensure independent measurements of the winds for future DWL missions."

6. What do you want to discuss using the HLOS winds projected onto the east-west and north-south directions in Figs. 7 and 8? What conclusions should the reader make from Figs. 7 and 8? In my opinion, the projected winds are not related to validation of Aeolus HLOS winds. The authors do not mention results obtained from Figs. 7 and 8 in Sect. 4. Please add some further explanations on that.

   Thank you for these questions. Regarding the decomposition in Figures 7 and 8, the decomposition into different wind-component directions provides insight into understanding the meteorological conditions that the measurements are sampling, which might be helpful to better understand the dynamical characteristics of this data in both Aeolus and other products. We have slightly modified the text to better explain this analysis (line 375):

   "Furthermore, some ascending and descending HLOS wind measurements cancel in the average owing to simply to the change of the angle of the LOS. To

avoid this artefact and to add some insight into the wind features being measured, we also compare the projected HLOS wind vector into its zonal (positive to the east) and meridional (positive to the north) components. The distribution of the zonal-component of the HLOS winds is shown in Fig. 7e and g for Aeolus and ECCC-B HLOS winds. By doing this decomposition, the distributions for ascending and descending measurements are brought into better agreement (Fig. 7f). We also notice that the HLOS winds can provide some information about the vertical variation of the HLOS winds that are projected onto the zonal direction (Figs. 7e and g). For example, for Aeolus the projection of HLOS into the zonal direction for the stratosphere, UTLS, and troposphere are +11.00 ms$^{-1}$, +4.00 ms$^{-1}$ and +1.00 ms$^{-1}$ respectively for this measurement period and these values (and the standard deviations of their distributions, see the figure legend for values) agree very well with ECCC-B (and ERA5 – not shown). The distributions have mean values that are positive because the winds are mainly westerly over the Arctic in the winter."

Regarding Figure 8, first, please note that, as stated above under point 1, Fig. 8 is now showing the distributions of the differences between the products.

The following sentences are added in the discussion section (Line 495):

"In our analysis of the pan-Arctic region, we found an overall agreement by comparing the distributions of the HLOS winds, ascending and descending HLOS winds, and projections of HLOS winds onto east-west and north-south directions in different atmospheric layers (Fig. 7), and we also compared the distributions of the differences between Aeolus and ECCC-B and ERA5 (Fig. 8). Due to the angle of the HLOS, when comparing the distributions, separating the ascending with descending measurements helps avoid cancelling out part of the HLOS winds and projecting the HLOS winds on to zonal and meridional directions provides some insight on the vertical variation of the HLOS winds."

7. Figs. 10 and 11 show spatial distributions of RMSD of Aeolus and ECCC-B vertical HLOS wind profiles. Figs. 10 and 11 are the most important result of the paper. The spatial distributions of RMSD show remarkable radial patterns. How do the authors explain these patters? Are these patterns due to interpolation of RMSD data to the grid points? The authors were not careful to ensure that the graphics all use similar color scales. Please use the same color scales for Figs. 10a–d and Figs. 11a–d. Similarly, please use the same color scales for Figs. 10e–h and Figs. 11e–h.

Thank you very much for highlighting this important issue and for the suggested improvements to the presentation. We found that the radial pattern was a spurious result arising from our choice of grids. We corrected this by transforming our data to the EASE (Equal-area scalable earth) grid, described at the NSIDC website (https://nsidc.org/data/ease). It is now corrected, and the following explanations are added in Line 436:

"Since the measurement density differs depending on the latitude, the RMSD of the profiles are calculated over nearly equal surface area, using the Equal-Area Scalable Earth (EASE) Grids (Brodzik et al., 2012). Each grid cell is around 104 km2 which is approximately the square of the along-path resolution of Aeolus Rayleigh winds."

Panels in Figs. 10 and 11 are now sharing the same colorbar.

8. Lines 446-447: Why no significant improvement is seen here? The estimated HLOS errors of the 2B10 data are decreased compared to the 2B06 data (Figs. S2 and S3). I cannot understand the authors' explanation "because we have implemented a weekly updated dynamic bias correction to the near real time data". Please add some further explanations on that.

Thank you for pointing this out. It is true that the estimated errors are decreased in the reprocessed data. We wanted to make a point that the same improvement is not seen in the O-B statistics.
The following sentences are added for clarification in Line 470:

"The estimated observational errors have decreased compared to the 2B06 data (Figs. S1 and S2) since the bias due to the M1 mirror temperature dependence is updated on a daily basis and the dark current signals have been removed using improved quality control. However, we do not see the same improvement in the O-B statistics between 2B06 and 2B10 products over the Arctic region."

**Specific comments**

1. Lines 23-24: "scattering from the solar background" should be revised to "the solar background radiation" or "the solar background noise". Thank you, fixed.

2. Line28: "all cases" should be revised to "all three periods". Thank you, fixed.

3. Line 28: "20%" should be revised to "5 to 40%". Thank you, fixed.

4. Lines 68-69: Please clarify what is meant by "new technologies" and "cost-effective alternatives to atmospheric monitoring".

   Thank you for the comment.

   "this project serves to test new technologies and provide cost-effective alternatives to atmospheric monitoring over the northern regions"

   is revised to

   "this project serves to test the spaceborne DWL that provides alternative observational wind data to atmospheric monitoring over the northern regions".

5. Lines 78-80: "Section 3.1 describes the comparison during the early FM-A period (15 September to 16 October 2018) to ground-based measurements in Canada's North, including the Iqaluit supersite and radiosonde stations over the Northern Canada." However, I can see the comparison results during the early FM-B and mid-FM-B periods in Fig. 5. Please correct.

   Thank you for pointing this out, the phrase "during the early FM-A period (15 September to 16 October 2018)" is removed.

6. Line 82: "1 December to 31 January 2020" should be revised to "1 December 2019 to 31 January 2020". Thank you, fixed.

7. Line 107: An Aeolus observation can be regarded as an averaged value of a 90 km line for the Rayleigh winds and Mie winds until 5 March 2019, and as an averaged value of a 10 km line for the Mie winds after 5 March 2019 (Martin et al. 2021). Please correct.

   Thank you for pointing this out. The passage (line 104) is revised to

   "Prior to 5 March 2019, both Rayleigh and Mie winds were averaged to up to a horizontal resolution of 87 km. Recognizing that Mie scattering in cloudy air yields stronger returns than Rayleigh scattering in clear air, after 5 March 2019, the Mie wind product was provided at a finer horizontal resolution of 12 km.".

8. Lines 115-116: In my opinion, the phrase "winter 2020" is a bit misleading. I would advise to avoid this phrase and rather use "winter 2019–20". Thank you, fixed.

9. Line 127: Please give the values of the thresholds for L2B estimated HLOS errors of Rayleigh-clear winds and Mie-cloudy winds.

Thank you for the comment. The following passage is added in line 125:

"The thresholds for L2B estimated observation errors during the FM-A period are 4.5 ms-1 for the Mie winds and 6.6 to 11 ms-1 for the Rayleigh winds, depending on the pressure level, and 5 ms-1 for the Mie winds and 8.5 to 12 ms-1 for the Rayleigh winds during the early FM-B period. For more details, please refer to Rennie and Isaksen (2020)."

10. Line 139: "ECCC-B background" should be revised to "ECCC-B". Thank you, fixed.

11. Lines 186-189 and 207-211: The authors downloaded the radiosonde data from http://weather.uwyo.edu/upperair/sounding.html. The vertical resolution of the radiosonde data is coarser than 15 m. The information of the geographical location and time at each level is not included in the radiosonde data. Is the balloon drift taken into account? Please given detailed data matching procedures between Aeolus and radiosonde.

Thank you for pointing this out. The raw radiosonde data is measured every 2 s, which results in a profile vertical resolution of 8-15 m. However, the data used from http://weather.uwyo.edu/upperair/sounding.html is the processed radiosonde data provided at standard pressure levels. It has a much coarser resolution than 15 m. The following passage is added in Line 191 for clarification:

"Vaisala RS92 radiosondes (Mariani et al., 2018) were launched twice daily (45 minutes before synoptic times 00 and 12 Coordinated Universal Time (UTC)). They measure vector wind profiles with a vertical resolution of roughly 15 m depending on ascent speed, up to about 30 km above ground level. The data used (available at http://weather.uwyo.edu/upperair/sounding.html) is the processed radiosonde data provided at mandatory and significant pressure levels (which has a coarser resolution than 15 m). It takes about two hours to reach 30 km altitude (around 10 hPa). The instrumental uncertainty for the wind speed is between 0.4 and 1.0 ms$^{-1}$ and between 0.3 and 0.7 ms$^{-1}$ for the zonal wind component (Dirksen et al., 2014). The error on the zonal wind component due to drift and elapsed time of the ascending balloon is between 0.5 and 1.0 ms$^{-1}$ in the troposphere and UTLS (see Fig.5b in Laroche and Sarrazin, 2013). As a result, the total error for the zonal wind component from these sources of errors is between 0.6 and 1.2 ms$^{-1}$. Note that the radiosonde data are assimilated in the ECCC and ECMWF systems, which means that the ECCC-B and ERA5 errors are not independent of the radiosonde observation errors. The ECCC Whitehorse site, situated in a wide valley with large lakes, also has radiosondes that operate similarly to the ones at the Iqaluit.".

Section 2.5 on the data matching process and coincidence criteria is also revised. "For the ground-based validation, the criterion for coincidence of Aeolus overpasses is that the distance from the sites to the measurements be no more than 90 km (horizontal resolution of Rayleigh winds). Using this coincidence criterion, Aeolus overpasses are selected as targets for validation at Iqaluit three times a week at around 21:50, 11:15, and 22:00 UTC, and at Whitehorse twice a week at around 02:25 and 15:30 UTC. The Aeolus measurements are compared to the reanalysis and in-situ measurements that are available in the nearest time. Temporal sampling for each product is as follows: Aeolus overpasses at Iqaluit and Whitehorse are as mentioned above; reanalysis data is provided hourly, on the hour; radiosonde data is from launches at 00 and 12 UTC, with a two-hour time-of-flight to 30 km as mentioned above; Ka-band radar data is provided via 15-minute scans. For example, if Aeolus overpasses selected as a target for validation at the Iqaluit site at 11:15 UTC, since the reanalysis data is sampled hourly, the radiosondes are launched at 00 and 12 UTC, and the Ka-band radar at Iqaluit scans every 15 minutes, the Aeolus HLOS profile would be compared to the reanalysis data and radiosonde measurements at 12 UTC and to the nearest scan by the radar. On the other hand, if the overpass time is 02:25 UTC, the profile would be compared to the ERA5 data at 02 UTC, the radiosonde measurements at 00 UTC, and, again, the nearest scan by the radar.".

12. Line 217: "overpasses Asia around 06 and 18 UTC" is error and needs to be corrected.

   This sentence is removed, and the paragraph is revised as mentioned above.

13. Figure 2: Please add y-axis title. Why are there two Rayleigh winds at the same altitude above 5 km in Fig. 2b. Similarly, why are there two Mie winds at about 8 km in Fig. 2a. Is this consistent with the temporal criterion described in lines 214-215. Why are the maximum altitudes of HLOS wind profile obtained from ECCC-B 15 km (20 km) in Fig. 2a (2b)? "at Iqaluit" should be added in the caption.

   There were two Aeolus measurements at the same level due to the collocation criteria. Figure 2 is now revised and is only showing the nearest profile from Aeolus to the sites.

   This is because the background values were provided at the observation locations. ECCC-B is linearly interpolated to Aeolus measurement locations and times.

The following passage is added in line 143 for clarification:
"The data used to compare with Aeolus winds in this paper is the assimilated data that is linearly interpolated to Aeolus measurement locations and times. For the linear interpolation between the model's grid points, the horizontal grid-spacing is 15 km and the vertical grid-spacing varies from approximately 100 m in the PBL to 1 km in the stratosphere (McTaggart-Cowan et al., 2019). The linear interpolation in time is between two consecutive model states, 15 min apart."

14. Figure 3 and Line 249: Is "frequency distributions in percentage" correct? What are the color shading areas in the scatter plots (Figs. 3a-d)?

The "frequency distributions in percentage" is correct. They are shown in the panels above and to the right of the scatter plots.

Line 268: "Figures 3 and 4 show scatter plots between the different datasets with lines of best fit and their range," is added.

15. Line 270: Is "The ERA5 shows somewhat slightly lower correlation" correct?

Table 1 is added. It shows the adjusted r-squared and slope of the fitted line for the in-situ comparison.

The paragraph in line 291 is revised to:
"Overall, the datasets show strong consistency. ECCC-B and ERA5 are highly mutually consistent (Table 1; with adjusted r-squared greater than 0.97) and therefore show similar consistency with Aeolus (Figs. 3a-b and 4a-b). It can be seen that Aeolus Mie winds are less consistent with ECCC-B, ERA5, and radiosondes at Iqaluit than the corresponding observations at Whitehorse and for the Rayleigh winds. One possible reason for this relates to the fact that the Mie channel samples winds in the lower atmosphere where winds are harder to assimilate or measure due to topography. Since Iqaluit is situated in tundra valleys with rocky outcrops that can cause increased variability in the wind field while Whitehorse is situated in large valleys with less wind variability due to topography, terrain effects might account for the difference in consistency. In addition, the overall range extent of the HLOS wind samples is between -25 to 25 ms$^{-1}$ at Iqaluit and -45 to 45 ms$^{-1}$ at Whitehouse and r-squared is sensitive to the range of data (note the denominator of the second term in Eq. (5)). Overall, Aeolus data show good agreement with these three datasets with adjusted r-squared greater than 0.8.".

16. Figure 4: Figure 4 is exactly the same as Fig. 3. Please correct. Thank you, fixed.

17. Figure 5: Why did the adjusted r-squared of Mie winds decrease with time? The authors should mention the decrease and discuss the reasons.

   Thank you for the question. Please note that the range on the y-axis is from 0.7 to 1.0. The difference may look large on the plot, but the change is almost insignificant. The 99% confidence level on the adjusted r-squared is added on the figure. The range of the adjusted r-squared for Mie winds is almost overlapping between the seasons. The following paragraph is added in Line 337 for clarification:

   "We also note a slight drop in consistency of the Mie winds for the mid-FM-B period, which took place in winter 2020: for instance, the adjusted r-squared and their 99% confidence intervals, between Mie winds and ECCC-B, are $0.92\pm0.03$ during fall 2018, $0.91\pm0.01$ during summer 2019, and $0.87\pm0.02$ during winter 2020. This decrease in the consistency is almost insignificant."

18. Line 321: "The reprocessed data has improved estimated errors and RMSD" should be revised to "The reprocessed data has improved estimated errors and RMSD over the excluded region". Thank you, fixed.

19. Figure 6: Please add y-axis titles. "Aeolus L2B estimated error" should be revised to "Aeolus L2B estimated error of Rayleigh winds". Thank you, fixed.

20. Line 335: "free troposphere (2-8 km)" should be revised to "free troposphere (T, 2-8 km)". Thank you, fixed.

21. Line 336: "stratosphere (altitude greater than16 km)" should be revised to "stratosphere (S, altitude greater than16 km)". Thank you, fixed.

22. Figure 7: Please add y-axis titles. Fig. 7. Figures. 7b and 7d do not use the same horizontal axis scale as Figs. 7f and 7h. Please correct. "70N" should be revised to "70°N". "each level" should be revised to "each atmospheric layer". Thank you, fixed.

23. Figure 8: Please add y-axis titles. Thank you, fixed.

24. Figure 9: Please add (a), (b), (c), (d), (e), and (f) to the image. Thank you, fixed.

25. Line 392: What do you mean with "more structure"? Please add some further explanations on that.

   Thank you for the comment. The sentence in line 411 is revised to:

   "Figure 9 shows that Aeolus data consistently has greater standard deviations than ECCC-B during all three periods and for both Rayleigh and Mie winds: its normalized standard deviations are typically within 1.05 to 1.40.".

26. Line 433: "observation errors" should be revised to "estimated errors of Rayleigh winds". Thank you, fixed.

**Technical corrections**

1. British or European English: For example, you use "16 October 2018" and "October 16th 2018".

2. Line 268: Fig. -> Figs.

3. Line 345: Fig. -> Figs.

4. Line 427: Fig. -> Figs.

5. Line 428: Fig. -> Figs.

6. Line 445: Fig. -> Figs.

7. Line 469: Fig. -> Figs.

Thank you, they are fixed now.

---

## Author Comment (AC2)

Referee comment on "Validation of the Aeolus Level-2B wind product over Northern Canada and the Arctic" by Chih-Chun Chou et al., Atmos. Meas. Tech. Discuss., https://doi.org/10.5194/amt-2021-247-RC2, 2021

We thank the reviewers for their careful reading of the manuscript and comments, which helped us improve the study.

**General comments:**

The authors compared the Aeolus wind measurements with Ka-radar, wireless soundings, ECCC, and ERA5 reanalysis data, and the results show that the Aeolus wind measurements are in good agreement with other wind measurements or reanalysis data. This paper has important implications for the application of Aeolus wind measurements in the Arctic, where wind measurements are currently scarce. However, the paper requires significant revision before publication, and the specific issues are described below.

1.  The authors use other soundings and reanalysis data as a benchmark for comparison with the Aeolus wind data, but the data quality of the other data is not presented in the paper. Perhaps the other wind data also have large biases in the Arctic, and then the authors' use of them as a comparison benchmark would make none of the comparisons in this paper credible. Especially for high-altitude wind fields, the data sets may have different data quality performances at different heights. Moreover, the reanalysis dataset itself contains assimilation of existing sounding data. The authors need to fully justify the reliability of the data in each of the datasets used in this paper.

    Thank you for pointing this out. To improve this aspect, we added a few more references on the validation of the datasets used in the study. For the radiosondes, Dirksen et al. (2014) described the uncertainty for the wind speed and Laroche and Sarrazin (2013) described the errors due to the radiosonde drifts. The following passage is added in Line 191 for clarification:

    "Vaisala RS92 radiosondes (Mariani et al., 2018) were launched twice daily (45 minutes before synoptic times 00 and 12 Coordinated Universal Time (UTC)). They measure vector wind profiles with a vertical resolution of roughly 15 m depending on ascent speed, up to about 30 km above ground level. The data used (available at http://weather.uwyo.edu/upperair/sounding.html) is the processed radiosonde data provided at mandatory and significant pressure levels (which has a coarser resolution than 15 m). It takes about two hours to

reach 30 km altitude (around 10 hPa). The instrumental uncertainty for the wind speed is between 0.4 and 1.0 ms$^{-1}$ and between 0.3 and 0.7 ms$^{-1}$ for the zonal wind component (Dirksen et al., 2014). The error on the zonal wind component due to drift and elapsed time of the ascending balloon is between 0.5 and 1.0 ms$^{-1}$ in the troposphere and UTLS (see Fig.5b in Laroche and Sarrazin, 2013). As a result, the total error for the zonal wind component from these sources of errors is between 0.6 and 1.2 ms$^{-1}$. Note that the radiosonde data are assimilated in the ECCC and ECMWF systems, which means that the ECCC-B and ERA5 errors are not independent of the radiosonde observation errors.".

As for the data quality for in-situ measurements, Mariani et al. (2018) demonstrates reliable observations from the Ka-band radar and its application, and Mariani et al. (2020) validates the ground-based lidar to the radiosondes. A few more sentences are added in line 209:

"The uncertainty of the measurements depends on conditions, SNR, and decibel of the return signal. The average vertical wind profile bias to radiosonde is better than 0.27 ms-1.".

Furthermore, it is true that ECCC-B and ERA5 contain assimilation of existing sounding data; however, they also assimilate many other observations, for instance, from the satellites, aircraft measurements, and more. Therefore, we think that they are still credible sources of wind data to validate Aeolus winds.

2. The authors' presentation of the spatio-temporal matching process between the datasets is not clear enough. It is suggested that the data matching process be introduced as a separate section and the information about the data used in this paper be summarized. A clear and detailed data matching process is desired to be seen.

Thank you for these suggestions. Although we considered introducing an additional section on this, we thought it would be best to revise Sect. 2.5 on the data matching and coincidence criteria. In doing this, we sought the same level of detail as provided by other publications including Belova et al. (2021), Martin et al. (2021), and Baars et al. (2020). Section 2.5 is revised to:

"For the ground-based validation, the criterion for coincidence of Aeolus overpasses is that the distance from the sites to the measurements be no more than 90 km (horizontal resolution of Rayleigh winds). Using this coincidence criterion, Aeolus overpasses are selected as targets for validation at Iqaluit three

times a week at around 21:50, 11:15, and 22:00 UTC, and at Whitehorse twice a week at around 02:25 and 15:30 UTC. The Aeolus measurements are compared to the reanalysis and in-situ measurements that are available in the nearest time. Temporal sampling for each product is as follows: Aeolus overpasses at Iqaluit and Whitehorse are as mentioned above; reanalysis data is provided hourly, on the hour; radiosonde data is from launches at 00 and 12 UTC, with a two-hour time-of-flight to 30 km as mentioned above; Ka-band radar data is provided via 15-minute scans. For example, if Aeolus overpasses selected as a target for validation at the Iqaluit site at 11:15 UTC, since the reanalysis data is sampled hourly, the radiosondes are launched at 00 and 12 UTC, and the Ka-band radar at Iqaluit scans every 15 minutes, the Aeolus HLOS profile would be compared to the reanalysis data and radiosonde measurements at 12 UTC and to the nearest scan by the radar. On the other hand, if the overpass time is 02:25 UTC, the profile would be compared to the ERA5 data at 02 UTC, the radiosonde measurements at 00 UTC, and, again, the nearest scan by the radar.".

3. the authors' analysis of Figure 5 is not rigorous enough and the conclusions drawn from Figure 5 are not reliable enough, see Specific comments L296.

   Please see the response below.

4. The authors need to clarify the practical significance of the discussion of the statistical distribution of the wind products themselves in Figures 7 and 8. It might be more valuable to discuss instead the distribution of the difference between the Aeolus data and the other data.

   Thank you for the suggestions. Regarding Fig. 7, we mainly want to show the Aeolus sampling in different atmospheric layers and that the decomposition into different wind-component directions provides insight into understanding the meteorological conditions that the measurements are sampling, which might be helpful to better understand the dynamical characteristics of this data in both Aeolus and other products. We have slightly modified the text to better explain this analysis (line 375):

   "Furthermore, some ascending and descending HLOS wind measurements cancel in the average owing to simply to the change of the angle of the LOS. To avoid this artefact and to add some insight into the wind features being measured, we also compare the projected HLOS wind vector into its zonal (positive to the east) and meridional (positive to the north) components. The

distribution of the zonal-component of the HLOS winds is shown in Fig. 7e and g for Aeolus and ECCC-B HLOS winds. By doing this decomposition, the distributions for ascending and descending measurements are brought into better agreement (Fig. 7f). We also notice that the HLOS winds can provide some information about the vertical variation of the HLOS winds that are projected onto the zonal direction (Figs. 7e and g). For example, for Aeolus the projection of HLOS into the zonal direction for the stratosphere, UTLS, and troposphere are +11.00 ms-1, +4.00 ms-1 and +1.00 ms-1 respectively for this measurement period and these values (and the standard deviations of their distributions, see the figure legend for values) agree very well with ECCC-B (and ERA5 – not shown). The distributions have mean values that are positive because the winds are mainly westerly over the Arctic in the winter."

Furthermore, we agree that it would be valuable to discuss the distribution of the difference between the Aeolus data and the other data. Thank you for the suggestion! Therefore, Fig. 8 now shows the means and standard deviations of the differences between Aeolus and ECCC-B and ERA5. The means of the differences therefore reflect the remaining bias between the datasets after the dynamic bias correction has been applied. The associated paragraph describing Fig. 8 is also revised. Starting at line 394:

"We compare the distributions of the differences between the Aeolus wind measurement data and the ECCC-B and ERA5 data during fall 2018, summer 2019, and winter 2020 over the Arctic, as summarized in Fig. 8, which shows the bias and standard deviations of the differences between Aeolus HLOS winds and the ECCC-B HLOS winds, and ERA5 HLOS winds, and their zonal and meridional projections. The measurements are decomposed into Rayleigh (red) and Mie winds (black). They are further decomposed into ascending (indicated with upright triangles) and descending (inverted triangles) measurements. The results, with the bias (the mean values of these differences for the different sampling used) being smaller than 0.7 ms$^{-1}$, are consistent with our bias correction method. The distributions of the differences in the ascending and descending measurements do not show a significant difference. The discrepancies in the meridional projections of the HLOS winds are smaller because Aeolus picks up mostly the zonal component of the winds due to the direction of the LOS."

**Specific comments:**

L82:1 December to 31 January 2020

Wrong time markings

Thank you, fixed.

L115: ECMWF has recently published the first reprocessed data (2B10)

Data links should be added after the data introduction.

Thank you, the link is added in Line 114: "ECMWF has published the first reprocessed data in fall 2020 (2B10; available at ftp://2018_aeolus_l2b:ecmwf@acquisition.ecmwf.int/), which covers the period between 24 June and 31 December 2019."

L125: The quality control recommendation following the Guidance for Aeolus NWP Impact Experiments (Rennie and Isaksen, 2019), including the threshold for L2B estimated observation errors.

The recommendation given by NWP is a threshold range. Authors should submit specific thresholds to be used when processing data.

Thank you for the comment. The threshold values are added in the text (line 125):

"The thresholds for L2B estimated observation errors during the FM-A period are 4.5 ms-1 for the Mie winds and 6.6 to 11 ms-1 for the Rayleigh winds, depending on the pressure level, and 5 ms-1 for the Mie winds and 8.5 to 12 ms-1 for the Rayleigh winds during the early FM-B period. For more details, please refer to Rennie and Isaksen (2020)."

L129: We further reject the outliers by excluding all the data when the difference between the observations and ECCC-B or ERA5 is greater than 30 m/s.

For the screening threshold of 30m/s, the authors need to give an explanation, either from data analysis or literature.

Thank you, the following explanation is added in Line 130: "This criterion was obtained from an initial comparison between Aeolus FM-A and ECCC-B (Laroche et al., 2019).".

L162: ECCC-B is then linearly interpolated to Aeolus measurement locations and times.

The process of linear interpolation needs to be clarified. In addition, the main comparison data in the latter section does not provide complete information on the vertical resolution, and due to the large number of datasets used in this paper, it is recommended to use a table after section 2.5 to summarize the important information of each dataset used.

Thank you for the comments. This passage is added in line 163 for clarification of the process of linear interpolation of ECCC-B:

"The data used to compare with Aeolus winds in this paper is the assimilated data that is linearly interpolated to Aeolus measurement locations and times. For the linear interpolation between the model's grid points, the horizontal grid-spacing is 15 km and the vertical grid-spacing varies from approximately 100 m in the PBL to 1 km in the stratosphere (McTaggart-Cowan et al., 2019). The linear interpolation in time is between two consecutive model states, 15 min apart."

We revised Sect. 2.5 (line 229) instead of adding a table after the section. It summarizes the temporal resolution of each dataset and clarifies the data matching process and coincidence criterion:

"For the ground-based validation, the criterion for coincidence of Aeolus overpasses is that the distance from the sites to the measurements be no more than 90 km (horizontal resolution of Rayleigh winds). Using this coincidence criterion, Aeolus overpasses are selected as targets for validation at Iqaluit three times a week at around 21:50, 11:15, and 22:00 UTC, and at Whitehorse twice a week at around 02:25 and 15:30 UTC. The Aeolus measurements are compared to the reanalysis and in-situ measurements that are available in the nearest time. Temporal sampling for each product is as follows: Aeolus overpasses at Iqaluit and Whitehorse are as mentioned above; reanalysis data is provided hourly, on the hour; radiosonde data is from launches at 00 and 12 UTC, with a two-hour time-of-flight to 30 km as mentioned above; Ka-band radar data is provided via 15-minute scans. For example, if Aeolus overpasses selected as a target for validation at the Iqaluit site at 11:15 UTC, since the reanalysis data is sampled hourly, the radiosondes are launched at 00 and 12 UTC, and the Ka-band radar at Iqaluit scans every 15 minutes, the Aeolus HLOS profile would be compared to the reanalysis data and radiosonde measurements at 12 UTC and to the nearest scan by the radar. On the other hand, if the overpass time is 02:25 UTC, the profile would be compared to the ERA5 data at 02 UTC, the radiosonde measurements at 00 UTC, and, again, the nearest scan by the radar."

L165: Reanalysis ERA5

For the ERA5 dataset, assimilation relationships between it and the other datasets used in this paper should be added.

Thank you. The following passage is added in Line 195 for clarification: "Note that the radiosonde data are assimilated in the ECCC and ECMWF systems, which means that the ECCC-B and ERA5 errors are not independent of the radiosonde observation errors.".

ERA5 only contains assimilation of radiosonde data and other observational data, but not the in-situ measurements used in the study. Therefore, even though ERA5 errors are dependent to the radiosonde observation errors, it is still worth comparing both datasets since ERA5 contains information from many other observations as well. An example of other study that uses both datasets is Chen et al. (2021).

Reference: Chen, S., Cao, R., Xie, Y., Zhang, Y., Tan, W., Chen, H., Guo, P., and Zhao, P.: Study of the seasonal variation in Aeolus wind product performance over China using ERA5 and radiosonde data, Atmos. Chem. Phys., 21, 11489–11504, https://doi.org/10.5194/acp-21-11489-2021, 2021.

L190-L210:

From the latter, there is very little overlap between Ka-band radar or LIDAR with Aeolus data. The main object of comparison is the radiosondes, so its data should have been presented in more detail. In particular, the horizontal drift problem of radiosondes, which seems to be not taken into account by the authors, may also lead to a large bias. In addition, is the introduction and use of Ka-band radar and ground-based LIDAR data in this paper meaningful? It seems that the absence of these two does not affect the logic and conclusions of this paper.

We acknowledge the little overlap between Ka-band radar or lidar with Aeolus data; however, they are still valuable because they are entirely independent from the Aeolus winds. Unlike the radiosondes, they are not assimilated in ERA5 and ECCC-B which are used in the correction of Aeolus winds and quality control process. Thus, the comparison would still add some value to this validation work. The retrieval of the radar wind profile is included because we have some comparison between Aeolus and radar in Figs. 3 and 4. The Aeolus wind is not compared to the ground-based is not

included, so its retrieval is not explained in the paper. However, since its measurements are shown in Fig. 2, a brief introduction to the instrument is still included.

As you mentioned, the horizontal drift problem of radiosondes has not been considered. So, we added more details on the data quality of radiosondes in Line 191 for clarification:

"Vaisala RS92 radiosondes (Mariani et al., 2018) were launched twice daily (45 minutes before synoptic times 00 and 12 Coordinated Universal Time (UTC)). They measure vector wind profiles with a vertical resolution of roughly 15 m depending on ascent speed, up to about 30 km above ground level. The data used (available at http://weather.uwyo.edu/upperair/sounding.html) is the processed radiosonde data provided at mandatory and significant pressure levels (which has a coarser resolution than 15 m). It takes about two hours to reach 30 km altitude (around 10 hPa). The instrumental uncertainty for the wind speed is between 0.4 and 1.0 ms$^{-1}$ and between 0.3 and 0.7 ms$^{-1}$ for the zonal wind component (Dirksen et al., 2014). The error on the zonal wind component due to drift and elapsed time of the ascending balloon is between 0.5 and 1.0 ms$^{-1}$ in the troposphere and UTLS (see Fig.5b in Laroche and Sarrazin, 2013). As a result, the total error for the zonal wind component from these sources of errors is between 0.6 and 1.2 ms$^{-1}$."

L246: On 24 September, Aeolus measures westerly winds in reasonable overall agreement with the other data.

The agreement between Aeolus and the other data in Figure 2b does not seem to be obvious, and the deviation in some data points is already close to 50%.

Thank you for pointing this out. We agree that the agreement is not that obvious.

The sentence (line 266) is revised: "On 24 September, although a few of the Rayleigh measurements have a deviation close to 50%, Aeolus still measures westerly winds in reasonable overall agreement with the other data.".

Figure 2 is also revised. There were two Aeolus measurements at the same level due to the collocation criteria. Figure 2 is now showing the nearest profile from Aeolus to the sites only.

L249:

Figures 3 and 4 are identical and need to be modified. The comparison of Ka-band radar in Figure 3d needs to be addressed for its significance since there are only about 10 data points. Also, the information on the fitted straight lines, standard deviation, and sample size in the figure can be added, while the information in the supplemental Figure S1 will be placed in the original text in the form of a table.

Thank you for pointing out this error and for the suggestions.

Figure 4 is now fixed and a F-test is performed as mentioned above (see response for L190-210).

Table 1 is added and Fig. S1 is removed (its information can be found in Table 1). Table 1 shows the adjusted r-squared and slope of the fitted line for the in-situ comparison.

L270:

The data consistency performance of the two sites is different, and the data consistency of ERA5 and ECCC-B with Aeolus in the Whitehorse site didn't change, and the conclusion given by the authors as well as the explanation is not reasonable enough.

Thank you for the comment. We agree that the data consistency performance of the two sites is different. We added some explanations on the differences. It is mainly due to the topography of the sites and the adjusted r-squared is sensitive to the wind speed range. The paragraph in line 291 is revised:

"Overall, the datasets show strong consistency. ECCC-B and ERA5 are highly mutually consistent (Table 1; with adjusted r-squared greater than 0.97) and therefore show similar consistency with Aeolus (Figs. 3a-b and 4a-b). It can be seen that Aeolus Mie winds are less consistent with ECCC-B, ERA5, and radiosondes at Iqaluit than the corresponding observations at Whitehorse and for the Rayleigh winds. One possible reason for this relates to the fact that the Mie channel samples winds in the lower atmosphere where winds are harder to assimilate or measure due to topography. Since Iqaluit is situated in tundra valleys with rocky outcrops that can cause increased variability in the wind field while Whitehorse is situated in large valleys with less wind variability due to topography, terrain effects might account for the difference in consistency. In addition, the overall range extent of the HLOS wind samples is between -25 to 25 ms$^{-1}$ at Iqaluit and -45 to 45 ms$^{-1}$ at Whitehouse and r-squared is sensitive to

the range of data (note the denominator of the second term in Eq. (5)). Overall, Aeolus data show good agreement with these three datasets with adjusted r-squared greater than 0.8.".

L275-L292:

The authors' discussion of Ka-band radar consistency with Aeolus and its causes seems unnecessary for this paper, as there are too few overlapping data points and no valid conclusions are drawn.

As mentioned above (see response for L190-210), we acknowledge the little overlap between Ka-band radar or lidar with Aeolus data; however, they are entirely independent from the Aeolus winds. Unlike the radiosondes, they are assimilated in ERA5 and ECCC-B, which is used in the correction of Aeolus winds and quality control process. Thus, the comparison would still add some value to this validation work.

L296:

Are the sample sizes in Figure 5 the same for the three time periods? The use of the expressions summer, fall, and winter is not rigorous and should be specific to dates. In addition, the discussion of Figure 5 is inadequate. If solar radiation is used to explain Figure 5, then why did the overall performance of fall 2018 be better than that of winter 2020? The Mie channel also performed better in summer 2019 than in winter 2020, and the Mie channel is also influenced by solar background radiation. The authors seem to have overlooked some phenomena in their haste to reach conclusions.

Thank you for the suggestions. Number of measurements (N) and number of profiles (p) are now provided in Table S1 to S3. Table S1 is for the validation at the sites, Table S2 for validation over the Canadian Arctic, and Table S3 for validation over the pan-Arctic. And the dates of the seasons are added to Fig. 5.

Thank you for the question. Please note that the range on the y-axis is from 0.7 to 1.0. The difference may look large on the plot, but the change is almost insignificant. The 99% confidence level on the adjusted r-squared is added on the figure. The range of the adjusted r-squared for Mie winds is almost overlapping between the seasons. The following passage is added in Line 337:

"We also note a slight drop in consistency of the Mie winds for the mid-FM-B period, which took place in winter 2020: for instance, the adjusted r-squared and their 99% confidence intervals, between Mie winds and ECCC-B, are 0.92±0.03 during fall 2018, 0.91±0.01 during summer 2019, and 0.87±0.02 during winter 2020. This decrease in the consistency is almost insignificant."

L355: Since ECCC-B and ERA5 are mutually consistent

This may need to be more fully demonstrated.

Thank you for pointing this out. We have tried make this consistency clearer with the following revision in line 291:

"Overall, the datasets show strong consistency. ECCC-B and ERA5 are highly mutually consistent (Table 1; with adjusted r-squared greater than 0.97) and therefore show similar consistency with Aeolus (Figs. 3a-b and 4a-b)."

*L359-L364:*

The conclusion of this paragraph does not need to be obtained by data analysis. It is just a mathematical law. The discussion of the longitudinal and latitudinal components also seems to be unnecessary.

Thank you for the comment. We agree with your point. Thus, this paragraph is removed.

Figure 8:

What is the significance of discussing the mean and standard deviation of the wind speed samples themselves? Clarification by the authors is needed. It may be more meaningful to discuss the distribution of the differences between the Aeolus wind measurement data and the ECCC-B and ERA5 data.

Please see above. Figure 8 is now showing the distributions of the differences.

L377: Figure 8 shows an overall agreement between Aeolus, ECCC-B, and ERA5

The proof of consistency between data by comparing data distribution characteristics only is not enough.

Thank you for the comments. Please see above. Figure 8 is now showing the distributions of the differences.

L391: Figure 9 shows that Aeolus data consistently has more structure than ECCC-B during all three periods and for both Rayleigh and Mie winds.

What does "more structure" mean here? Please explain.

Thank you for this question, by "more structure" we mean that the Aeolus' normalized standard deviations are greater than 1.0 for all three seasons and for both channels, and we explain this in the following revision in line 411:

"Figure 9 shows that Aeolus data consistently has greater standard deviations than ECCC-B during all three periods and for both Rayleigh and Mie winds: its normalized standard deviations are typically within 1.05 to 1.40.".

L395: During the boreal summer period, the data in the stratosphere seem to agree less with the ECCC-B data, reflecting reduced sampling, solar background noise that is most effective during summer as mentioned earlier, and other possible errors (Reitebuch et al., 2020).

The derivation of this conclusion is not rigorous, there are many possible reasons for the decrease in the correlation between the Aeolus data and ECCC-B data in the stratosphere in summer, and it is also possible that it is caused by changes in the atmospheric environment in summer, thermal changes in the telescope, etc. I don't think we can make speculation on the cause from Figure 9. But the authors seem to attribute it to solar radiation in the abstract.

Thank you for the comment. In the text, we mentioned that there are other possible errors, but we only mentioned the solar background noise in the abstract – this is our mistake. We added in the abstract that the decrease in the consistency for Rayleigh winds in the summer may be due to other possible errors as well.

L402: For this reason, in the next paragraph, where we investigate the spatial distribution of the consistency in the lower and upper atmospheric regions, we exclude the Rayleigh winds in the PBL and the Mie winds in the stratosphere.

It would have been more convincing if the authors had made this data trade-off from the analysis of data used in this paper. Although Aeolus was designed to complement the dual channels at altitude, there are many cases where the Mie channel has higher data volume and data quality at altitude than at lower altitudes, especially in summer. Simply removing the data would be detrimental to the subsequent analysis.

Thank you for the comment. The choice of removing the Rayleigh winds in PBL and Mie winds in the stratosphere mainly comes from Fig. 9. There are two seasons out of the three seasons of study that there is no Rayleigh measurement in the PBL or the normalized standard deviation is greater than 2.2, and one season that there is no Mie measurement in the stratosphere or with normalized standard deviation greater than 2.2. The Rayleigh winds in the PBL are really noisy from the Taylor diagrams and from Rennie and Isaksen, 2020. Therefore, for the consistency across the periods of study, we remove the layer that provides very little and/or noisy data.

Figure10, Figure11:

The calculation process of RMSD in this paper should be clearer, and it is better to give the formula and the range of RMSD to be considered "consistent". In addition, the radial mutations in these two figures seem to be inconsistent with common sense, especially in Figure 11e, which I hope the authors can explain. Also, the same color scale should be used for all subplots. The number of data samples used for different subplots should be provided, because the valid sample size may vary greatly for different seasons at different altitudes. In addition, the data density of Aeolus is also different for different latitudes, how did the authors deal with this point, and does this lead to lower quality data for lower latitudes?

Thank you very much for highlighting this important issue and for the suggested improvements to the presentation. This was very helpful! We have learned that the radial pattern was a spurious result arising from our choice of grids. We corrected this by transforming our data to the EASE (Equal-area scalable earth) grid, described at the NSIDC website (https://nsidc.org/data/ease). It is now corrected, and the following explanations are added in Line 436:

"Since the measurement density differs depending on the latitude, the RMSD of the profiles are calculated over nearly equal surface area, using the Equal-Area Scalable

Earth (EASE) Grids (Brodzik et al., 2012). Each grid cell is around 104 km$^2$ which is approximately the square of the along-path resolution of Aeolus Rayleigh winds."

Panels in Figs. 10 and 11 are now sharing the same colorbar.

L446: No significant improvement is seen here because we have implemented a weekly updated dynamic bias correction to the near real time data.

How do the authors explain the slightly higher RMSD of 2B10 data compared to 2B06 in Figure 10 and Figure 11? What is the meaning of the dynamic bias correction mentioned in the paper, please elaborate, and what is the difference between this correction and the reprocessed data correction in the L2B product?

Thank you for pointing this out. We realized that the averaged RMSD shown in the subtitles are dependent on the grids chosen. Since we are now using different grids (EASE 2) (please see response for Figs. 10 and 11), the averaged RMSD also changed. To avoid misleading results, the average is removed from the subtitles and the paragraph in line 469 is revised:

"Note that the first reprocessed data, 2B10, only overlaps with one of the three periods of study: August to September 2019. The estimated observational errors have decreased compared to the 2B06 data (Figs. S1 and S2) since the bias due to the M1 mirror temperature dependence is updated on a daily basis and the dark current signals have been removed using improved quality control. However, we do not see the same improvement in the O-B statistics between 2B06 and 2B10 products over the Arctic region.".

L476: We have found some initial evidence that the estimated error product is also a good predictor of RMSD between Aeolus and the reanalysis, which could be useful for constraining future forecasts.

The "estimated error product" itself is used to estimate the difference between the Aeolus wind product and the true wind field. If it does not predict the RMSD between Aeolus and reanalysis data, then the reanalysis data deviates from the real wind field. I don't understand the purpose of the author's statement, perhaps more information about "constraining future forecasts" is needed.

We agree that this sentence should be clarified. What we meant by constraining future forecasts is that the assimilation of Aeolus winds, in an optimal way, would improve analyses and forecasts. Consequently, the analyses and forecasts would be closer to the true atmospheric state. One way to get the best from assimilating Aeolus winds

into forecasting systems is to use most accurate observation errors for these data. We found that the L2B estimated error product could be useful for specifying the HLOS wind observation errors since there is a good correspondence between the spatial variability of the time-averaged L2B estimated error products (Figs. S1 and S2) and that of the RMSD between Aeolus and ECCC-B (Figs. 10 and 11). The sentence has been changed as follows:

"We found that the spatial variability of the time-averaged L2B estimated error product is in good agreement with the spatial variability of the RMSD between Aeolus and ECCC-B HLOS winds over the Arctic region. This validates the use of L2B estimated error product as a predictor for the HLOS wind observation errors in data assimilation systems, as proposed by Rennie et al. (2021), to obtain optimal positive impacts on forecasts from assimilating Aeolus winds."

---

## Referee Report (RR1)

**Comment on amt-2021-247**

**General comments**

The manuscript presents a comparison between Aeolus observations in the Arctic and corresponding ground-based remote observations (using a Ka-band radar and a Doppler lidar), in-situ observations from radiosondes and model output from ECCC-B and ERA5.

The scope of the comparison is to assess the consistency between Aeolus and the other wind sources and to validate its L2B error product in a region where direct wind observations are especially sparse.

The analysis of the authors shows that there is good agreement between Aeolus observations and radiosonde measurements and between Aeolus observations and model output from ECCC-B and ERA5. The differences between Aeolus and ECCC-B also correspond for the most part to the errors found in the Aeolus L2B product. The authors also confirm the issue, already shown in previous studies, of Aeolus Rayleigh faulty observations in summertime conditions due to contamination from solar background radiation.

Despite being the second time this manuscript has been submitted for revision, after taking into account the comments of the anonymous reviewers in the first round, it still presents deficiencies that do not make it ready for publication.

For what concerns the ground-based measurements (Ka-band radar and Doppler lidar), their amount is too limited in order to make a comparison with a good statistical significance. I therefore suggest to either increase the amount of observations or to completely remove them from the analysis of this manuscript and leave the focus on the comparison with radiosondes and model output.

Moreover, there are many issues throughout the manuscript that need to be corrected and clarified, as specified by the following comments.

**Specific comments**

- Line 34: correct "surfaced-based" with "surface-based".
- Line 163: I guess that for the comparison with Aeolus you used the analysis data (output of 4D-EnVar) rather than the assimilated data (input of 4D-EnVar, i.e. the observations).
- Lines 164-167: what is the interpolation sequence? First horizontal, then vertical, then temporal?
- Line 195: define the acronym UTLS.
- Figure 1: choose a different aspect ratio to make the figure more squared, change the combination of colors (magenta over red doesn't have a good contrast) and increase the image resolution (the red dots are blurred) to improve readability.
  The circles, rather than being circles of different size, should resemble ellipses of different elongation. If you are not able to plot ellipses (I realize that it's not an easy task), it's better to just plot them as equal-sized dots to show the location of the different sites.
- Lines 213-214: specify the three unknown parameters.
- Line 215: the ftp site is not accessible because a password is required.
- Lines 216-221: why bothering including the Doppler lidar instrument in this study if its data will not be used eventually? The visual comparison of Figure 2 brings little information as the vertical range is limited to 3 km.

- Line 236: *"if Aeolus overpasses selected as a target for validation at the Iqaluit site at 11:15 UTC"*: I am missing the verb in this sentence. Perhaps you forgot to add something like "occur at 11:15 UTC"?
- Lines 236-238: *"since the reanalysis data is sampled hourly, the radiosondes are launched at 00 and 12 UTC, and the Ka-band radar at Iqaluit scans every 15 minutes"*: there is no need to repeat this information, it is already provided in the lines above (lines 234-236).
- Line 238: why Aeolus at 11:15 UTC is compared to reanalysis data at 12 UTC and not to reanalysis data at 11 UTC, which is the closest in time?
- Lines 239-240: when do the scans by the radar exactly occur? At HH:00, HH:15, HH:30, HH:45 (where HH is the hour)? If it is so, you can also specify the exact time of the radar, for example 11:15 UTC for the first example and 02:30 UTC for the second example.
- Line 246: add 2018 to 22 September.
- Line 249: it is the opposite, HLOS winds are negative westward during the ascending phase. HLOS wind is defined to be positive when blowing away from the instrument and the instrument is pointing eastward during the ascending phase.
- Line 250: *"we plot the profile of negative HLOS winds to ease the interpretation"*: this sentence is unclear. Do you mean that you invert the sign of the whole (ascending phase) profile to match the sign of the descending phase profile (or better the other way around, since in the caption of Figure 2 you say that the zonal projection is positive eastwards, which corresponds to an ascending profile)?
- Figure 2: it would help to have a background grid corresponding to the tick marks of the x and y axis. In the caption the "zonal component" should be called "zonal projection". The radar profile extends to less than 5 km, however there are Aeolus Mie measurements around 8 km (left panel), meaning that at that altitude there could be clouds and therefore hydrometeors detectable by the radar. How do you explain this lack of observation by the radar? Did you find a similar behavior in the other profiles too?
- Line 271: remove "during fall 2018" because this condition (more Rayleigh measurements than Mie) is valid for all the periods, not only fall 2018.
- Figures 3 and 4: increase the image resolution.
  Choose another color combination because blue and black dots do not have a good contrast next to each other.
  What is the utility of plotting the frequency distributions?
  What do the colored bands around the best fit lines represent?
  In the caption replace "(a) background" with "(a) ECCC-B".
  I guess that the radar measurements come from Iqaluit only, as you state in the text. Looking at the number of measurements reported in Table S1, there are 60 measurements matched with Aeolus Rayleigh at Iqaluit, but in Figure 3(d) there are only about 11-12 points. Similarly, Table S2 reports only 12 measurements matched with Aeolus Mie but in Figure 4(d) there are much more than 12 points.
  Furthermore, as the radar measurements are associated with more cloudy conditions, I would expect to have more profiles from the Mie channel matched with radar profiles compared to Rayleigh, as the Mie channel is associated with cloudy conditions as well. However in Table S1 it is the opposite.
- Line 274: add "the" before "consistency".
- Lines 275-276: center the equations.
- Line 278: replace "$\bar{y}$ is the mean of $y$" with "$\bar{y}$ is the mean of $y_i$".

- Line 280: in the following lines, only the values of the adjusted r-squared of Table 1 are discussed, but there is no comment about the values of the slope of the fitted line.
- Lines 291-294: *"It can be seen that Aeolus Mie winds are less consistent with ECCC-B, ERA5, and radiosondes at Iqaluit than the corresponding observations at Whitehorse and for the Rayleigh winds."*: I would rewrite this sentence like this: "It can be seen that Aeolus winds are in general less consistent with ECCC-B, ERA5, and radiosondes at Iqaluit than the corresponding observations at Whitehorse. Moreover, at Iqaluit, Rayleigh winds show a higher consistency than Mie winds, while the opposite is true for Whitehorse."
- Line 298: replace "Whitehouse" with "Whitehorse".
- Lines 297-299: the denominator of the second term of Eq. 5 doesn't have anything to do with the wind range. N is the number of measurements and p the number of profiles.
- Table S1: how come there are Ka-band radar measurements at Whitehorse if you stated in the main text that such radar is installed only at Iqaluit?
- Line 310: replace "distinctive" with "distinct" or "different".
- Figure 5: add background horizontal grid lines.
- Line 340: the decrease in the consistency for Mie in winter 2020 doesn't look that insignificant. Can you try to explain what could be the cause of that decrease?
- Line 350: after "9 September" add "indicated by the vertical red dashed line".
- Line 350: perhaps it is worth adding that the decrease in estimated errors in this particular period shows how the contribution to the error due to the solar background radiation is decreasing with the transition from summer to fall conditions.
- Line 352: "During this period": which period are you exactly referring to? 2 August to 30 September 2019, only September 2019 or the day of the jump (9 September)?
- Line 353: replace "averaging" with "averaged".
- Line 354: replace "on" with "along".
- Lines 355-356: *"Thus, as a trade-off of having high vertical resolution, the Aeolus estimated errors are larger for this specific range bin setting."*: I would rewrite this sentence like: "Thus, the price for having a higher vertical resolution is larger errors for this specific range bin setting."
- Line 371: replace "2km" with "2 km".
- Line 375: replace "cloudy condition" with "cloudy conditions".
- Figure 7: the distributions on the left appear to be "stacked" distributions. If this is the case, it must be declared both in the caption and in the main text, otherwise they might be confused with overlapping distributions and interpreted differently.
  Increase the image resolution.
  How come you show the HLOS u-projection for Rayleigh only? What about the HLOS u-projection for Mie?
- Lines 372-374: *"Rayleigh winds are more frequently sampled in the UTLS and the stratosphere since often cloud layers are too optically thick for the laser to penetrate (an example distribution for winter 2020 over the Arctic is shown in Fig. 7a)."*: this sentence is not entirely correct because besides the effect of clouds on limiting the view of the Rayleigh channel beyond cloud layers you also have to consider that the UTLS and the stratosphere layers simply span a larger vertical range (8 to 30 km) compared to the PBL and the free troposphere (0 to 8 km), hence there are more observations for those layers. In order to estimate the number of observations for each atmospheric layer it would be better to compute the height distribution of the observations instead of the HLOS wind distribution.

Furthermore, I find it a bit strange that for Rayleigh there are no observations at all in the PBL: can you please check your data again?

- Lines 374-375: *"The Mie channel measures winds under cloudy condition and thus has more measurements in the PBL than in the stratosphere (e.g., Fig. 7c)."*: don't forget the free troposphere. I would change the sentence like this: "The Mie channel measures winds under cloudy condition and thus has more measurements in the PBL and in the free troposphere than in the upper layers (e.g., Fig. 7c)."

- Line 376: "to simply" remove "to".

- Lines 375-377: *"Furthermore, some ascending and descending HLOS wind measurements cancel in the average owing to simply to the change of the angle of the LOS."*: I would replace this sentence with: "Furthermore, the ascending and descending Rayleigh distributions (Fig. 7b) are symmetric about zero due to the symmetric azimuth angle of the instrument with respect to the north when switching from the ascending to the descending phase.

- Line 379: replace "for Aeolus" with "for Aeolus Rayleigh" and add "Rayleigh" also in the corresponding part of the figure caption.

- Lines 380-382: *"We also notice that the HLOS winds can provide some information about the vertical variation of the HLOS winds that are projected onto the zonal direction (Figs. 7e and g)."* I would rewrite this sentence as follows: "We also notice that the projected zonal component of the HLOS winds can provide some information about the vertical variation of the zonal wind."

- Lines 382-385: *"For example, for Aeolus the projection of HLOS into the zonal direction for the stratosphere, UTLS, and troposphere are +11.00 ms$^{-1}$, +4.00 ms$^{-1}$ and +1.00 ms$^{-1}$ respectively for this measurement period and these values (and the standard deviations of their distributions, see the figure legend for values) agree very well with ECCC-B (and ERA5 – not shown)."* I would rewrite this sentence as follows: "For example, for Aeolus Rayleigh the mean values of the zonal projection of the HLOS wind for the stratosphere, UTLS and troposphere are 11.00 ms$^{-1}$, 4.00 ms$^{-1}$ and 1.00 ms$^{-1}$ respectively. These mean values, as well as their standard deviations (see legend of Figs. 7e and g), agree well with ECCC-B (and ERA5 – not shown)."

- Lines 385-386: *"The distributions have mean values that are positive because the winds are mainly westerly over the Arctic in the winter."* Please cite a reference about the statistics of winds in the Arctic.

- Lines 394-397: *"We compare the distributions of the differences between the Aeolus wind measurement data and the ECCC-B and ERA5 data during fall 2018, summer 2019, and winter 2020 over the Arctic, as summarized in Fig. 8, which shows the bias and standard deviations of the differences between Aeolus HLOS winds and the ECCC-B HLOS winds, and ERA5 HLOS winds, and their zonal and meridional projections."* This sentence is too repetitive, modify the second part like this: "We compare the distributions of the differences between the Aeolus wind measurement data and the ECCC-B and ERA5 data during fall 2018, summer 2019, and winter 2020 over the Arctic, as summarized in Fig. 8, which shows the bias and standard deviations of the differences for the HLOS winds and for their zonal and meridional projections.

- Line 397: replace "are decomposed" with "are separated".

- Line 397: replace "red" with "red dots".

- Line 398: replace "black" with "black dots".

- Line 399: replace "bias" with "biases".

- Line 399: remove "the mean values of these differences for the different sampling used", there is no need to explain what the bias is.
- Line 400: replace "are consistent with our bias correction method" with "are consistent with ECCC bias correction method".
- Lines 400-401: *"The distributions of the differences in the ascending and descending measurements do not show a significant difference."* To be correct, here you are not showing the distributions but the means and standard deviations of the distributions. You can have different distributions (in shape) with equal mean and standard deviation.
- Lines 401-402: *"The discrepancies in the meridional projections of the HLOS winds are smaller because Aeolus picks up mostly the zonal component of the winds due to the direction of the LOS."* The vertical range of Figure 8 (-7 to +7 m/s) doesn't allow to grasp the subtle differences of the biases around zero. We can only see the differences between standard deviations. By "discrepancies" do you mean the standard deviations?
- Figure 8: increase image resolution.
  Perhaps, to highlight the variations of the means near zero, you could reduce the vertical scale to [-1 1] or [-2 2] and divide the standard deviations by a factor 10 (specifying this change in the text and in the caption).
- Line 409: replace "to the star" with "the distance to the star".
- Lines 452-453: *"The RMSD are systematically greater in the lower-atmosphere than in the upper-atmosphere as shown in Figs. 10 and 11"*: it seems the opposite, the RMSD is smaller in the lower atmosphere (Fig. 10) than in the upper atmosphere (Fig. 11).
- Line 453: what is the estimated error product? The errors contained in the Aeolus L2B product?
  Figures S1 and S2 have no caption so you should better explain what they represent in the text.
- Figure 10: how do you explain the increase in the error in (d) for longitudes between 60° W and 120° E?
- Lines 472-473: *"However, we do not see the same improvement in the O-B statistics between 2B06 and 2B10 products over the Arctic region."* Could you provide a reason why the improvement is not seen?
- Figure 11: how do you explain the large anisotropy in (e)?
  You explained that the large errors in Summer for Rayleigh are due to contamination of solar background radiation. How do you explain that the errors are still high in Winter for Rayleigh (d)? Such errors are in fact lower in the L2B error product (Fig. S2d).
- Line 477: replace "the Aeolus data product are bias corrected" with "the Aeolus data products are bias-corrected".
- Line 490: add "2019" after "1 December".
- Lines 501-502: *"the standard deviations of Aeolus winds are 5 to 40% greater than ECCC-B in every layer."*: a standard deviation 40% greater than the reference means a normalized standard deviation of 1.4, but in Figure 9 there are values above that.
- Lines 504-505: *"In any case, this analysis reveals consistent HLOS winds with correlations higher than 0.8 except during summer in the stratosphere and normalized standard errors within one standard deviation of ECCC-B."*: there are other layers and periods with correlation coefficients below 0.8: PBL in (a), (b) and (c) and S in (d). Also, there are normalized standard errors above one standard deviation in in (a), (b), (c), (d) and (e).

---

## Author Response (AR2)

(The line numbers refer to the line number showed on the revised manuscript with track changes.)

1. To address to Reviewer 3's concern that there were limited coincident measurements between Aeolus and Ka-band radar and ground-based Doppler lidar (comment Q1), the following passages have been revised:

   - Revisions to the abstract: At line 17, we state: "**In particular, Aeolus data is compared to a limited sample of coincident ground-based Ka-band radar measurements at Iqaluit, Nunavut**". At line 23, we state: "**except for lower values for the comparison with the Ka-band radar, reflecting limited sampling opportunities with the radar data**". At line 30, we add: "**Our work shows that the high quality of the Aeolus dataset that has been demonstrated globally applies to the sparsely sampled Arctic region. It also demonstrates the lack of available independent wind measurements in the Canadian North, lending urgency to the need to augment the observing capacity in this region to ensure suitable calibration and validation of future space-borne DWL missions.**"

   - In the methods, at line 223, the material the described the Doppler lidar at Iqaluit has been shortened and moved to the results section. In particular, at line 223 "Lastly, the Doppler lidar measures the LOS component of wind and, similarly to the radar, can retrieve horizontal winds via the VAD method. However, it is used only for visual comparison in this study (in the example profiles of Fig. 2) because it has very few coincident measurements with Aeolus due to its small vertical range, about 3 km a.g.l. or the cloud base height. Nevertheless, the Doppler lidar wind-profile observations were found to have measurements consistent with high-resolution radiosondes (Mariani et al., 2020), which should be borne in mind when considering our validation of Aeolus against radiosondes."

     has been revised to

     "**We will also briefly mention ground-based Doppler lidar measurements in Sect. 3.1.**" in line 193

     and "There are three" in line 191 has been revised to "**We focus on two**"

     and the corresponding passage has been added to the results in line 262:

     "**The Iqaluit site also hosts a ground-based Doppler lidar whose LOS wind measurements yields horizontal vector winds via VAD up to about 3 km a.g.l. or the cloud base height. This instrument provides very limited coincident measurement opportunities with Aeolus due to its limited vertical range. Observations from this instrument, which have been extensively validated against high-resolution radiosondes (Mariani et al., 2020) and are useful for boundary-layer focused work, are also shown in Fig. 1 for visual comparison only for 22 and 24 September and will not be considered further.**"

   - In the results section, at line 260: The passage

     "The Ka-band radar's vertical range extends to less than 5 km in both profiles, around where there are Mie wind measurements from Aeolus, because its vertical range depends

on hydrometeor concentration; the lidar's vertical range only extends to around 2 to 3 km. Due to the limited region of comparison, the agreement between Aeolus and radar is less good as we will discuss later, and we will not consider the lidar measurements further in this study."

is revised to

"**The Ka-band radar's vertical range extends to less than 5 km in both profiles, around where there are Mie wind measurements from Aeolus, because its vertical range depends on hydrometeor concentration, and its sampling of coincident timing is limited for the Aeolus measurement period.**"

- In the results section, at line 285: The passage and Fig.3 that compares between radiosonde and radar is added:
  "**Although the Ka band radar offers limited sampling, it is retained in this analysis because it offers an entirely independent and unique set of observations in the Canadian North that are not assimilated in any NWP model. Furthermore, it provides consistent measurements with the radiosondes, as shown in Fig. 3. For the same period of analysis and when the radar observations are within 30 minutes of radiosonde launch, the bias of the wind speed between the radar and radiosonde is less than 1 ms$^{-1}$ for measurements above 200 m a.g.l., and the standard deviation of the differences are within 3 ms$^{-1}$**"

- In the conclusion, at Line 536:

"The comparison with the Ka-band radar at Iqaluit has been limited to the early phase of Aeolus lifetime due to some technical issues from the ground-based radar. The agreement between Aeolus wind product and the Ka-band radar is systematically worse than with the forecasts and reanalysis products. We acknowledge the little overlap data with Aeolus due to the radar's limited sampling, but the comparison between Aeolus and radar, which are totally independent, is still at 99% confidence level using F-test. This provides encouragement for programs like CAWS to enhance independent radar measurements over Canadian Arctic sites."

has been revised as follows:

"**The comparison with the Ka-band radar at Iqaluit has been limited to the early phase of Aeolus lifetime due to a technical failure of the ground-based radar requiring extensive repair. The agreement between Aeolus wind product and the Ka-band radar is systematically worse than with the forecasts and reanalysis products. Nevertheless, for the limited sampling available, we have verified that the Ka-band radar provides consistent vector winds above 200 m a.g.l. with a bias less than 1 ms-1 and a RMSD less than 3 ms-1, and that the radiosonde measurements are consistent with Aeolus winds with an adjusted r-squared greater than 0.8. While the Ka-band radar data availability was limited, its analysis is retained in this paper because this independently generated ground-based data, which is unique in the large geographical region of the Canadian North, is not assimilated in any NWP system and is critical to validate Aeolus in this part of the Arctic. This highlights that radar observations are rare and challenging to obtain because of costs and**

**logistics and the sparsity of independently generated ground-based data in the Arctic. Because this critically limits validation capacity for this region, these results encourage programs like CAWS to enhance independent radar measurements over the Canadian Arctic and to continue investment in such infrastructure.**"

2. To address to Reviewer 3's comment Q3 and Q4, the description of the ECCC-B interpolation has been revised
Line 168: "The data used to compare with Aeolus winds in this paper is the assimilated data that is linearly interpolated to Aeolus measurement locations and times." has been revised to "**For the comparison between ECCC-B and Aeolus winds, the closest short-range forecast field, available every 15 minutes, is selected. Then, this field is linearly interpolated in space to Aeolus measurement locations, first horizontally and then vertically**"

3. To address to Reviewer 3's comment Q6, Fig. 1 has been revised:
   - The circles have now equal radius and "(some circles appear differently sized because of map-projection distortion)" in line 208 has been deleted.
   - The lon/lat limits have been changed.
   - The color for Iqaluit and Whitehorse sites is now green instead of magenta.

4. To address Reviewer 3's comments Q10 to Q12, the passage on description of coincident criterion has been revised.
Line 243: "For example, if Aeolus overpasses selected as a target for validation at the Iqaluit site at 11:15 UTC, since the reanalysis data is sampled hourly, the radiosondes are launched at 00 and 12 UTC, and the Ka-band radar at Iqaluit scans every 15 minutes, the Aeolus HLOS profile would be compared to the reanalysis data and radiosonde measurements at 12 UTC and to the nearest scan by the radar. On the other hand, if the overpass time is 02:25 UTC, the profile would be compared to the ERA5 data at 02 UTC, the radiosonde measurements at 00 UTC, and, again, the nearest scan by the radar."
has been revised to
"**For example, if Aeolus overpasses selected as a target for validation at the Iqaluit site occur at 11:15 UTC, the Aeolus HLOS profile would be compared to the reanalysis data at 11 UTC, to radiosonde measurements at 12 UTC, and to the nearest scan by the radar. On the other hand, if the overpass time is 02:25 UTC, the profile would be compared to the ERA5 data at 02 UTC, the radiosonde measurements at 00 UTC, and, again, the nearest scan by the radar.**"

5. To address Reviewer 3's comment Q15,
"When the measured HLOS winds are positive westward, i.e., when Aeolus is in its ascending orbit phase, we plot the profile of negative HLOS winds to ease the interpretation."
is revised to
"**When the measured winds are positive, it means the HLOS winds are directed away from the instrument (eastward for the ascending orbit phase and westward for the descending orbit phase). To ease the interpretation, we plot the negative HLOS winds when Aeolus is in its descending orbit phase (panel b).**"

6. To address Reviewer 3's comment Q18, "during fall 2018" in line 297 has been deleted.

7. To address Reviewer 3's comment Q23, "**All adjusted r-squared values in this comparison are above 0.95 for both sites and the slopes of the fitted line are all $1 \pm 0.1$.**" is added in line 310.

8. To address Reviewer 3's comment Q30, line 372: "This decrease in the consistency is almost insignificant" is revised to "**The cloud cover, number of observations, and estimated error**

**from Aeolus do not seem to control this decrease. Its cause, which could be due to the Aeolus measurement or the wind retrieval, remains unclear."**

9. To address Reviewer 3's comment Q32, we added this sentence in line 385: "**This decrease in this period also shows how the contribution to the error due to the solar background radiation is decreasing with the transition from summer to fall conditions**".

10. To address Reviewer 3's comment Q40, the sentence in line 409:
"Rayleigh winds are more frequently sampled in the UTLS and the stratosphere since often cloud layers are too optically thick for the laser to penetrate (an example distribution for winter 2020 over the Arctic is shown in Fig. 7a)"
has been revised to
"**Figure 8a and c show examples of stacked distributions of the Rayleigh and Mie winds for winter 2020 over the Arctic. Rayleigh winds at pressure greater than 850 hPa are ignored as recommended by Rennie and Isaksen (2020), because they show some indications of degradation in forecasts.**"

11. To address Reviewer 3's comment Q41, line 413
"The Mie channel measures winds under cloudy condition and thus has more measurements in the PBL than in the stratosphere (e.g., Fig. 7c)."
has been revised to
"**The Mie channel measures winds under cloudy conditions and thus has more measurements in the PBL and in the free troposphere than at higher altitudes (e.g., Fig. 8c).**"

12. To address Reviewer 3's comment Q42, line 415
"Furthermore, some ascending and descending HLOS wind measurements cancel in the average owing to simply to the change of the angle of the LOS."
has been revised to
"**Furthermore, the ascending and descending Rayleigh distributions (Fig. 8b) are symmetric about zero due to the symmetric azimuth angle of the instrument with respect to the north when switching from the ascending to the descending phase.**"

13. To address Reviewer 3's comment Q44, line 422
"We also notice that the HLOS winds can provide some information about the vertical variation of the HLOS winds that are projected onto the zonal direction (Figs. 7e and g)."
has been revised to
"**We also notice that the projected zonal component of the HLOS winds can provide some information about the vertical variation of the zonal wind.**"

14. To address Reviewer 3's comment Q45, line 424
"For example, for Aeolus the projection of HLOS into the zonal direction for the stratosphere, UTLS, and troposphere are +11.00 ms$^{-1}$, +4.00 ms$^{-1}$ and +1.00 ms$^{-1}$ respectively for this measurement period and these values (and the standard deviations of their distributions, see the figure legend for values) agree very well with ECCC-B (and ERA5 – not shown)."
has bee revised to
"**For example, for Aeolus Rayleigh, the mean values of the zonal projection of the HLOS wind for the stratosphere, UTLS and troposphere are 11.00 ms$^{-1}$, 4.00 ms$^{-1}$ and 1.00 ms$^{-1}$ respectively. These mean values, as well as their standard deviations (see legend of Figs. 8e and g), agree well with ECCC-B (and ERA5 – not shown).**"

15. To address Reviewer 3's comment Q46, line 427,
"The distributions have mean values that are positive because the winds are mainly westerly over the Arctic in the winter."

has been revised to

"**Aeolus-measured positive values of the zonal wind component from the stratosphere into the troposphere is consistent with the known climatological presence of westerlies in this region in polar winter. Analyzing the zonal projection of the HLOS winds highlights this feature**".

16. To address Reviewer 3's comments Q47 to Q56, Fig. 9's vertical scale has been reduced by dividing the standard deviations by a factor of 10. And the paragraph about it has also been revised.

    Line 438: "We compare the distributions of the differences between the Aeolus wind measurement data and the ECCC-B and ERA5 data during fall 2018, summer 2019, and winter 2020 over the Arctic, as summarized in Fig. 8, which shows the bias and standard deviations of the differences between Aeolus HLOS winds and the ECCC-B HLOS winds, and ERA5 HLOS winds, and their zonal and meridional projections. The measurements are decomposed into Rayleigh (red) and Mie winds (black). They are further decomposed into ascending (indicated with upright triangles) and descending (inverted triangles) measurements. The results, with the bias (the mean values of these differences for the different sampling used) being smaller than 0.7 ms-1, are consistent with our bias correction method. The distributions of the differences in the ascending and descending measurements do not show a significant difference. The discrepancies in the meridional projections of the HLOS winds are smaller because Aeolus picks up mostly the zonal component of the winds due to the direction of the LOS."

    has been revised to

    "**We compare the distributions of the differences between the Aeolus wind measurement data and the ECCC-B and ERA5 data during fall 2018, summer 2019, and winter 2020 over the Arctic, as summarized in Fig. 9, which shows the bias and standard deviations of the differences for the HLOS winds and for their zonal and meridional projections. To highlight the variations of the means, the standard deviations were divided by a factor of 10. The measurements are separated into Rayleigh (red dots) and Mie winds (black dots). They are further separated into ascending (indicated with upright triangles) and descending (inverted triangles) measurements. The results, with the biases being smaller than 0.7 ms-1, are consistent with ECCC bias correction method. The means and standard deviations of the differences in the ascending and descending measurements do not show a significant difference. The discrepancies in the meridional projections of the HLOS winds are smaller because Aeolus picks up mostly the zonal component of the winds due to the direction of the LOS**".

17. To address Reviewer 3's comment Q58 and 59, line 500
    "The RMSD are systematically greater in the lower-atmosphere than in the upper-atmosphere as shown in Figs. 10 and 11"
    has been revised to
    "**The RMSD are systematically greater in the upper-atmosphere than in the lower-atmosphere as shown in Figs. 11 and 12, and the differences could be anticipated from the Aeolus estimated errors from L2B product (as shown in Figs. S1 and S2)**".

18. To address Reviewer 3's comment Q61, "**, possibly because the reprocessed product had improved the precision (error characterization) of the measurements while not leading to a change in the overall agreement between the products, suggesting that accuracy is not changed**" has been added in line 522 after "Arctic region".

19. To address Reviewer 3's comment Q65, line 564,

"the standard deviations of Aeolus winds are 5 to 40% greater than ECCCB in every layer"
has been revised to
"**the standard deviations of Aeolus winds are 5 to 40% greater than ECCC-B in every layer with abundant Aeolus measurements, i.e., above the troposphere for Rayleigh winds and below the lower stratosphere for Mie winds, with an exception for the stratospheric Rayleigh winds in summer.**".

20. To address Reviewer 3's comment Q65, line 568
"In any case, this analysis reveals consistent HLOS winds with correlations higher than 0.8 except during summer in the stratosphere and normalized standard errors within one standard deviation of ECCC-B."
has been revised to
"**In any case, this analysis reveals consistent Rayleigh HLOS winds above the troposphere and Mie HLOS winds below the lower stratosphere with correlations higher than 0.8 except during summer in the stratosphere due to the solar background noise and normalized standard errors within one standard deviation of ECCC-B.**"

21. Some minor changes have been made on the wording:
   - Line 16: "progressing" has been inserted.
   - Line 16: "measurement stations" has been revised to "specific locations".
   - Line 40: "surfaced-based" has been revised to "surface-based".
   - Lines 66, 81, 82, and 206: "northern Canada" has been revised to "the Canadian North".
   - Line 201: "UTLS" has been revised to "upper troposphere/lower stratosphere (UTLS)".
   - Line 212: "cloud" has been deleted.
   - Line 215: "0.27" has been rounded up to "0.3".
   - Line 217: "In other words, it is scanning" has been revised to "whereby the radar scans".
   - Line 218: "and is known function of time" has been deleted.
   - Line 221: "($u$, $v$, and $w$)" has been added after "these three unknown parameters".
   - Line 254: "2018" has been added after "22 September".
   - Lines 274 and 275: "component" has been revised to "projection"
   - Line 276: "In Fig. 2, we see that" has been added.
   - Line 282: "from the other datasets around 8 km a.g.l.," has been added after "50%".
   - Line 290: Figure 3 has been added. In text, the figure number has been revised thereafter.
   - Line 296: "(ECCC-B, ERA5, radiosonde, and the limited number of coincident Ka-band radar profiles)" has been added.
   - Line 301: "the" is added before "consistency".
   - Line 305: "$y$" has been revised to "$y_i$".
   - Line 320: "Mie" has been deleted before "winds" and "in general" has been added before "less consistent".
   - Line 321: "and for the Rayleigh winds" has been removed and "Moreover, at Iqaluit, Rayleigh winds show a higher consistency than Mie winds, while the opposite is true for Whitehorse." has been added.
   - Line 327: "Whitehouse" has been revised to "Whitehorse".
   - Line 338: "distinctive" has been revised to "distinct".
   - Figures 4 and 5: Data at Whitehorse has been changed from blue to red.
   - Figure 6: background horizontal grid lines have been added.
   - Line 384: "indicated by the vertical red dashed line" has been added after "9 September".

- Line 387: "During this period" has been revised to "From 9 September to 6 October 2019".
- Line 389: "averaging" has been revised to "averaged".
- Line 390: "on" has been revised to "along".
- Line 392: "as a trade off of having higher vertical resolution, the Aeolus estimated errors are larger" has been revised to "the price for having a higher vertical resolution is larger errors".
- Line 419: "stacked" has been added in front of "distribution".
- Line 420: "Rayleigh" has been added after "Aeolus".
- Line 421: "decomposition" has been revised to "projection".
- Line 450: "Although" has been added in front of "Fig. 9" and "but" has been added in front of "more analysis".
- Line 455: "the distance" has been added in front of to the star.
- Line 459: "or greater" has been added after "1.05 to 1.40"
- Line 462: "for Rayleigh winds above the troposphere and for Mie winds below the lower stratosphere, with an exception for stratospheric consistency for Rayleigh winds during the summer." has been added after "greater than 0.8".
- Line 552: "2019" has been added after "1 December".